# Single-cell transcriptomes of developing and adult olfactory receptor neurons in *Drosophila*

Colleen N McLaughlin[1†], Maria Brbić[2†], Qijing Xie[1,3], Tongchao Li[1], Felix Horns[4,5], Sai Saroja Kolluru[4,6], Justus M Kebschull[1], David Vacek[1], Anthony Xie[1], Jiefu Li[1,7], Robert C Jones[4], Jure Leskovec[2], Stephen R Quake[4,6,8*], Liqun Luo[1*], Hongjie Li[1]

[1]Department of Biology, Howard Hughes Medical Institute, Stanford University, Stanford, United States; [2]Department of Computer Science, Stanford University, Stanford, United States; [3]Neurosciences Graduate Program, Stanford University, Stanford, United States; [4]Department of Bioengineering, Stanford University, Stanford, United States; [5]Biophysics Graduate Program, Stanford University, Stanford, United States; [6]Chan Zuckerberg Biohub, Stanford, United States; [7]Biology Graduate Program, Stanford University, Stanford, United States; [8]Department of Applied Physics, Stanford University, Stanford, United States

**Abstract** Recognition of environmental cues is essential for the survival of all organisms. Transcriptional changes occur to enable the generation and function of the neural circuits underlying sensory perception. To gain insight into these changes, we generated single-cell transcriptomes of *Drosophila* olfactory- (ORNs), thermo-, and hygro-sensory neurons at an early developmental and adult stage using single-cell and single-nucleus RNA sequencing. We discovered that ORNs maintain expression of the same olfactory receptors across development. Using receptor expression and computational approaches, we matched transcriptomic clusters corresponding to anatomically and physiologically defined neuron types across multiple developmental stages. We found that cell-type-specific transcriptomes partly reflected axon trajectory choices in development and sensory modality in adults. We uncovered stage-specific genes that could regulate the wiring and sensory responses of distinct ORN types. Collectively, our data reveal transcriptomic features of sensory neuron biology and provide a resource for future studies of their development and physiology.

***For correspondence:**
steve@quake-lab.org (SRQ);
lluo@stanford.edu (LL)

[†]These authors contributed equally to this work

**Competing interests:** The authors declare that no competing interests exist.

## Introduction

Detection of sensory stimuli is critical for animals to find food, identify mates, recognize suitable habitats, and evade predators and harmful conditions. Specialized sensory neurons have evolved to discriminate and transmit information relating to chemical, thermal, and hygrosensory stimuli. In *Drosophila*, the ~58 types of primary sensory neurons that detect these cues are found in the third segment of the antenna, a branched structure emanating from the antenna called the arista, and the maxillary palp (*Figure 1A*). The majority of neurons in the antenna are olfactory receptor neurons (ORNs) that respond to a variety of volatile compounds (*Hallem et al., 2004*; *Hallem and Carlson, 2006*; *Silbering et al., 2011*). A subset of neurons in the antenna and arista respond to temperature and humidity-related stimuli (*Yao et al., 2005*; *Ni et al., 2013*; *Barbagallo and Garrity, 2015*; *Enjin et al., 2016*; *Knecht et al., 2017*). Each of the ~52 types of antennal sensory neurons expresses a distinct sensory receptor, or a unique combination of 2–3 receptors (*Couto et al., 2005*; *Fishilevich and Vosshall, 2005*; *Goldman et al., 2005*; *Silbering et al., 2011*). Neurons that express

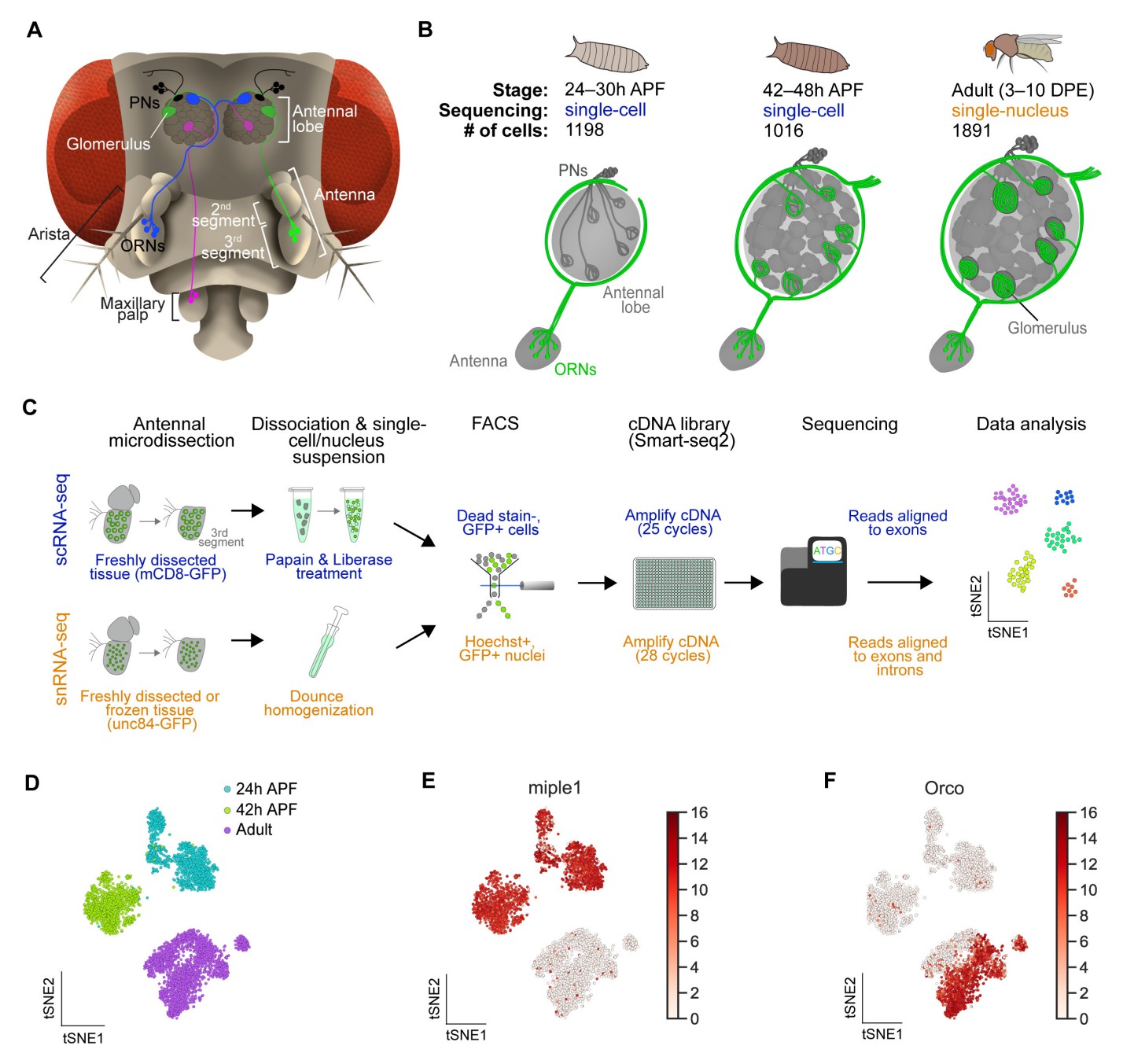

**Figure 1.** Single-cell transcriptomic profiling of *Drosophila* olfactory receptor neurons. (A) Schematic of the *Drosophila* olfactory system. Three types of olfactory receptor neurons (ORNs) are portrayed in three different colors (green, magenta, and blue) and their cell bodies are housed in the antennae and maxillary palps. Axons of a given ORN type form synapses with dendrites of a given type of projection neuron (PN; black) in the antennal lobe of the brain in discrete glomeruli. (B) Diagram containing information about the olfactory circuit and sequencing performed at three stages: 24–30 hr APF (h APF; hours after puparium formation), 42–48 hr APF, and adults (3–10 DPE; days post-eclosion). Each ORN type expresses a unique olfactory receptor or a unique combination of receptors and sends its axons to a stereotyped glomerulus. 42 hr APF ORNs were previously sequenced in *Li et al., 2020*. (C) Diagram highlighting key differences in single-cell (blue) and single-nucleus (orange) RNA sequencing protocols. Cells are labeled with membrane-bound mCD8-GFP, whereas nuclei are marked with the nuclear envelop-bound unc84-GFP. (D) t-distributed stochastic neighbor embedding (t-SNE) plot depicting all sequenced ORNs. Dimensionality reduction was performed using the top 416 differentially expressed (DE) genes from ORNs at all stages. (E–F) Expression of genes found in developing ORNs (E) and a subset of mature ORNs (F). Heat map units, $\log_2(\text{CPM}+1)$ (CPM: counts per million reads).

The online version of this article includes the following figure supplement(s) for figure 1:

*Figure 1 continued on next page*

*Figure 1 continued*

**Figure supplement 1.** Filtering and quality control of sequenced ORNs.

the same receptor(s) also project their axons to the same glomerulus of the antennal lobe in the brain (*Couto et al., 2005*; *Fishilevich and Vosshall, 2005*; *Benton et al., 2009*; *Silbering et al., 2011*). Here, their axons form one-to-one connections with the dendrites of second-order projection neurons (PNs), thus creating discrete and anatomically stereotyped information-processing channels (*Figure 1A*). For simplicity, we will broadly refer to the neurons within the third antennal segment as ORNs because they are the vast majority of neurons in this structure and their development and function are better studied, even though a small number of hygrosensory and thermosensory neurons are also present (*Barbagallo and Garrity, 2015*).

Prior to carrying out their function in the adult, ORNs undergo multiple steps of development. Following their birth in the early pupal stages, ORNs extend their axons from the antenna into the brain where they take distinct trajectories around the antennal lobe, a process that positions them closer to their glomerular targets (*Joo et al., 2013*; *Li et al., 2018*; *Figure 1B*, left). Axons next enter the antennal lobe to find and synapse with their partner PN's dendrites (*Jefferis et al., 2004*; *Mosca and Luo, 2014*; *Figure 1B*, middle). Concomitant with circuit formation and maturation, all ORN types express their unique olfactory receptor(s). Both the stereotyped development and organization of this system make it an excellent model to study molecular mechanisms of neuronal specification, axon guidance, and sensory physiology. Previously, bulk RNA sequencing of the whole antenna has been used to investigate gene expression patterns during development and in the adult (*Menuz et al., 2014*; *Li et al., 2016*; *Barish et al., 2017*; *Mohapatra and Menuz, 2019*); however, the cell-type-specific transcriptional changes that underlie ORN development and function remain elusive.

Single-cell RNA sequencing (scRNA-seq) is a powerful tool to investigate the mechanisms underlying nervous system development and function (*Ofengeim et al., 2017*; *Mu et al., 2019*; *Li, 2020*). Recent scRNA-seq studies in the *Drosophila* nervous system have enabled the discovery of new cell types, mechanisms important for development and aging, and genes controlling neural specification and wiring (*Li et al., 2017*; *Croset et al., 2018*; *Davie et al., 2018*; *Brunet Avalos et al., 2019*; *Allen et al., 2020*; *Kurmangaliyev et al., 2020*; *Özel et al., 2021*). Our recent study at a mid-developmental stage demonstrates that single-cell transcriptomic clusters represent anatomically and functionally distinct ORN types (*Li et al., 2020*). Yet, the mechanisms enabling these distinct sensory neuron types to transition through discrete developmental states into functional adult cells remain enigmatic. Furthermore, it has been difficult to obtain single-cell transcriptomic access to adult ORNs as these cells, like many other peripheral tissues in insects, are encapsulated within the hardened cuticle and cannot be isolated without substantial damage.

Here, we developed a single-nucleus RNA-seq (snRNA-seq) protocol to profile adult ORNs, thermosensory and hygrosensory neurons. We also used single-cell RNA-seq to profile these neurons at an early pupal stage. On a global level, we observed a shift in gene expression as neurons advance through development, from axon guidance-related transcripts to those important for sensory function. At the cell-type level, transcriptomic clusters corresponded to anatomically and functionally defined neuron types, allowing us to assign the cell-type identity to ~60% of developing and ~80% of adult clusters. Cell-type-specific transcriptomes, in part, reflected axon trajectory choice in early development and physiological functions in adults. Finally, we uncovered a large number of genes that could regulate circuit assembly during development and the detection or transmission of sensory information in the adult stage. Altogether, this dataset provides an important foundation for investigating molecular mechanisms controlling sensory neuron development and function.

## Results

### Single-cell transcriptomic profiling of *Drosophila* olfactory receptor neurons

Recently, we performed scRNA-seq on ORNs at a mid-developmental timepoint, 42–48 hr after puparium formation (APF; hereafter 42 hr APF), as their axons are in the final stage of selecting

synaptic partner PNs (*Li et al., 2020*; *Figure 1B*, middle). We sought to more completely understand how transcriptomic changes underlie the differentiation and function of these cells by profiling them at two additional stages: 24–30 hr APF and 3–10 day old adult (hereafter 24 hr APF and adult, respectively). At 24 hr APF, ORN axons have just begun to circumnavigate the antennal lobe after selecting a specific trajectory, which is necessary for proper glomerular targeting (*Figure 1B*, left) (*Joo et al., 2013*). Thus, we reasoned that scRNA-seq of 24 hr APF neurons would provide us with a more comprehensive picture of their development and axon targeting. Adult single-cell transcriptomes are important for understanding how distinct ORN types detect and transmit sensory information, and adult cell types should be readily distinguishable by the specific sensory receptors they express. However, adult ORNs are emmeshed in the hardened cuticle, making it difficult to isolate them without substantial loss of RNA quality using standard scRNA-seq dissociation methods.

Therefore, we adapted single-nucleus RNA-seq (snRNA-seq) to profile adult ORNs (*Figure 1C*), as similar methods have been used to gain transcriptomic access to cell types that are resistant to standard dissociation protocols in other organisms (*Habib et al., 2017*; *Denisenko et al., 2020*; *Ding et al., 2020*; *Kebschull et al., 2020*). Our snRNA-seq protocol is comparable to our scRNA-seq method, in which GFP-labeled neurons in the third antennal segment were dissected and dissociated into a single-cell suspension, sorted into 384-well plates via fluorescence activated cell sorting (FACS), and sequenced using the SMART-seq2 platform (*Picelli et al., 2014*; *Figure 1C*, top). snRNA-seq differed from scRNA-seq at four steps (*Figure 1C*, bottom). First, instead of labeling neuronal membranes with *mCD8-GFP*, we labeled their nuclear envelopes with *unc84-GFP* (*Henry et al., 2012*). Second, nuclei were dissociated mechanically rather than enzymatically. Third, nuclei were labeled with Hoechst, a fluorescent DNA stain, prior to being sorted into 384-well plates via FACS. Finally, since nuclei contain less RNA content than whole cells, nuclear cDNA was amplified using three additional PCR cycles and sequencing reads were aligned to both exons and introns, instead of exons alone for scRNA-seq (*Figure 1C*; see *Figure 2*).

By combining scRNA-seq and snRNA-seq, we sequenced 24 hr APF and adult ORNs to a depth of ~1 million reads per cell/nucleus. Following quality filtering for cells/nuclei with $\geq$ 50,000 reads that express at least two out of five neuronal markers (see Materials and methods), we obtained 1,198 and 1,891 high-quality neurons at 24 hr APF and adult, respectively (*Figure 1—figure supplement 1A*). Together with the 1,016 ORNs from 42 hr APF (*Li et al., 2020*), we profiled a total of 4,105 neurons, representing three distinct developmental stages (*Figure 1B*). Confirming transcriptomic quality, we observed high expression of neuronal markers, and the antennal cell marker *pebbled* (*peb*), but did not detect the glial marker *reversed polarity* (*repo*) in cells at all three stages (*Figure 1—figure supplement 1B*). Transcriptomes could be readily separated into their respective developmental stages following unbiased clustering using differentially expressed genes identified from all stages (*Figure 1D*). We observed stage-specific expression of select genes such as *miple1* in 24 hr and 42 hr APF ORNs (*Figure 1E*), which has known expression in the developing central nervous system (*Celniker et al., 2009*). Further, we detect adult-specific expression of the olfactory co-receptor *Orco* (*Larsson et al., 2004*; *Figure 1F*). These data demonstrate the reliability of our single-cell and single-nucleus RNA-seq protocols and suggest that substantial transcriptomic changes occur as development proceeds.

## Comparison of single-nucleus and single-cell RNA-seq

Prior to examining ORN transcriptomes, we compared our snRNA-seq protocol to scRNA-seq. Since we were unable to directly evaluate adult ORN cells and nuclei, we sequenced 24 hr APF ORN nuclei to compare with cells at the same stage (*Figure 2A*). To increase the number of cells analyzed, we combined the transcriptomes of 24 hr APF ORN nuclei labeled by driving nuclear-GFP with either the pan-neuronal *elav-GAL4* or *AM29-GAL4* which labels two ORN types (*Figure 2A*; *Figure 2—figure supplement 1A*; *Endo et al., 2007*). We used pan-neuronal *nSyb-GAL4* to label adult ORNs. To evaluate if this technique was broadly applicable to fly neurons, we also sequenced adult PN nuclei using *VT033006-GAL4* to label the majority of PN types (*Tirian and Dickson, 2017*; see the companion manuscript *Xie et al., 2021* for details) and compared them with stage-matched cells labeled with *GH146-GAL4* (*Figure 2A*).

We began by comparing the number of uniquely mapped reads and genes detected in cells and nuclei. We expected that snRNA-seq reads would be lower than scRNA-seq because nuclear transcripts are only a fraction of those found in the entire cell. When we aligned sequencing reads to

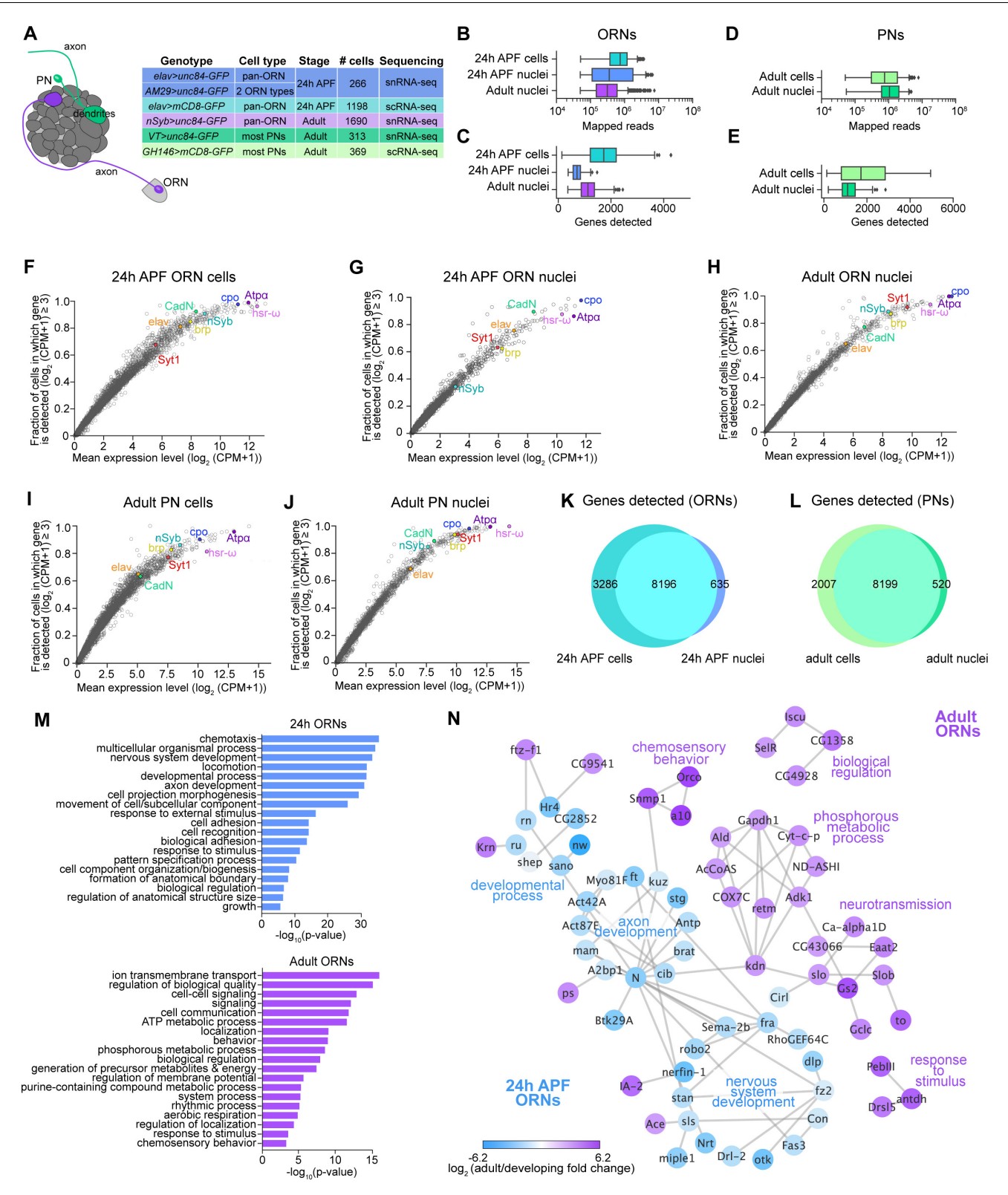

**Figure 2.** Comparison of single-cell and single-nucleus RNA-seq. (**A**) Schematic depicting ORN axons and PN dendrites in the antennal lobe (left) and summary table of the cell types, cell numbers, genotypes, and stages that were profiled using snRNA-seq and scRNA-seq. Note that 24 hr APF cells were labeled by intersecting *elav-GAL4* with *ey-FLP* and *UAS-FRT-STOP-FRT-mCD8-GFP*. The reason for fewer adult nuclei here than in **Figure 1B** is because some adult nuclei were labeled with lam-GFP, but those nuclei were not included in analyses here (see **Figure 6—figure supplement 1F**). (**B**–

*Figure 2 continued on next page*

*Figure 2 continued*

E) Box plots depicting uniquely-mapped reads (B, D) and detected genes (C, E) per cell/nucleus. Only exonic reads are shown for RNAs in cells, whereas exonic and intronic reads are combined for nuclear transcripts. (F–J) Mean expression level and detection rate of all genes in 24 hr APF ORN cells (F), 24 hr APF ORN nuclei (G), adult ORN nuclei (H), adult PN cells (I), and adult PN nuclei (J). Detection failure events can occur due to (1) the gene not being expressed in the cell; (2) gene dropouts resulting from technical artifacts even though mRNA transcripts are present; or (3) gene expression is below detection threshold. Colored circles represent the five neuronal markers [ *embryonic lethal abnormal vision* (*elav*); *neuronal Synaptobrevin* (*nSyb*); *Synaptotagmin 1* (*Syt1*); *Cadherin-N* (*CadN*); *bruchpilot* (*brp*)] used for quality control filtering (***Figure 1—figure supplement 1B***) and three additional genes [*couch potato* (*cpo*); *Heat shock RNA ω* (*hsr-ω*); *Na+ pump α subunit* (*ATPα*)]. (K–L) Venn diagram comparing all genes detected using scRNA-seq and snRNA-seq in 24 hr APF ORNs (K) and in adult PNs (L). Expression was defined as log$_2$(CPM +1) ≥ 2. (M) Gene Ontology (GO) analysis based on the top ~365 genes enriched in 24 hr ORN nuclei compared to adult ORN nuclei (top) and GO analysis of top ~365 genes enriched in adult ORN nuclei compared to 24 hr APF nuclei (bottom). Top 19 GO terms are shown. (N) STRING plot showing gene networks of 24 hr APF and adult ORN nuclei.

The online version of this article includes the following figure supplement(s) for figure 2:

**Figure supplement 1.** Additional comparisons of scRNA-seq and snRNA-seq.

exons only, both the number of mapped reads and genes detected in ORN nuclei were reduced compared to cells (***Figure 2—figure supplement 1B,C***). We tested if aligning nuclear sequencing reads to exons and introns could increase the gene detection rate, since intronic reads comprised 32.7% and 26.2% of total reads in 24 hr APF and adult ORN nuclei, respectively, compared to 10.5% when included in 24 hr APF cell reads (***Figure 2—figure supplement 1D***). Indeed, inclusion of intronic reads increased average gene detection rate in nuclei (***Figures 2B, C Figure 2—figure supplement 1C***). Similarly, intronic reads were 27.4% of total PN reads; when they were added into exonic reads, the mean number of genes detected increases from ~1,000 to ~1,150 in nuclei (***Figure 2D,E***; ***Figure 2—figure supplement 1E***). Thus, including intronic reads in snRNA-seq data provides additional information that is not captured by exonic reads alone.

We next assessed gene expression variability between cells and nuclei. Changes in gene expression can arise from biological factors, such as differences in transcription rate or cell state, or are the result of technical artifacts leading to missed gene detection or 'gene dropouts' (***Li and Xie, 2011***; ***Munsky et al., 2012***; ***Little et al., 2013***). We evaluated expression of the five neuronal markers used for quality filtering of cells and nuclei (*elav, CadN, Syt1, brp, nSyb*) and additional widely expressed genes (*cpo, Atpα, hsr-ω*) to determine the contribution of gene dropouts to expression variability. The majority of genes we evaluated showed similar expression in both ORN and PN cells and nuclei (***Figure 2F–J***). However, some genes, namely *nSyb* and *brp*, had a higher dropout rate in 24 hr nuclei compared to cells at the same stage (***Figure 2F,G***), a finding that is in line with a previous report in mammalian neurons (***Bakken et al., 2018***). Overall, gene dropouts likely do not contribute to considerable expression differences between cell and nuclear transcriptomes.

Next, we evaluated the extent to which snRNA-seq and scRNA-seq can detect similar genes. We first assayed the 20 genes with highest expression by percentage of total reads and found that stage-matched cells and nuclei shared a subset of the same highly expressed genes (***Figure 2—figure supplement 1F vs. G, I vs. J*** in red). As expected, cells showed higher detection of mitochondrial transcripts than nuclei (***Figure 2—figure supplement 1F,I***). Second, we compared the total genes detected in cells and nuclei. More than 70% of genes detected in ORN or PN cells were found in their nuclear counterparts, and more than 90% of nuclear genes were detected in cells (***Figure 2K–L***), indicating that snRNA-seq can detect similar genes to scRNA-seq. Finally, we queried the types of genes that were differentially expressed between cells and nuclei by performing Gene Ontology (GO) analysis between stage-matched cell and nuclear transcriptomes (***Ashburner et al., 2000***). Based on studies in other species (***Bakken et al., 2018***; ***Deeke and Gagnon-Bartsch, 2020***), we predicted that cells would be enriched for genes with house-keeping functions, probably resulting from preferential cytoplasmic mRNA stability. Indeed, cellular transcriptomes were enriched for genes in GO categories such as translation, protein folding, and metabolic processes, whereas nuclear transcriptomes contained terms related to stage-specific processes (***Figure 2—figure supplement 1K,L***). Most of the stage-specific genes were also expressed in ORN cells, albeit at slightly reduced levels relative to nuclei (***Figure 2—figure supplement 1N***, last column), indicating that our snRNA-seq and scRNA-seq protocols detect mostly overlapping genes. As expected, RNAs

expressed from the mitochondrial genome were enriched in ORN cells, whereas long non-coding RNAs were enriched in nuclei (*Figure 2—figure supplement 1M,N*).

We note two instances where we detect potentially artifactual gene expression in our adult ORN nuclei. First, some mitochondrial transcripts were found in adult nuclei (*Figure 2—figure supplement 1H*). Mitochondrial transcripts are among the highest detected genes in our single-cell transcriptomes (*Figure 2—figure supplement 1F,I*) but not in 24 hr APF ORN or PN nuclei (*Figure 2—figure supplement 1G,J*). Second, transcripts encoding odorant-binding proteins (Obps), known to be expressed at extremely high levels in accessory antennal cells (*Menuz et al., 2014*; *Larter et al., 2016*), were detected in adult ORN nuclei (*Figure 2—figure supplement 1O*). Detection of mitochondrial and Obp transcripts in our adult nuclei was likely from contamination during isolation of nuclei from the hardened cuticle.

Finally, we asked if snRNA-seq could reveal transcriptomic changes associated with neuronal maturation. Thus, we computed the differentially expressed genes between nuclei at 24 hr APF and the adult stage and identified GO terms for each timepoint. Developing (24 hr APF) transcriptomes were enriched for terms that describe axon and tissue development, cell adhesion, and cell recognition (*Figure 2M,N*). Adult specific-genes, on the other hand, were involved in processes like signaling, cell communication, and chemosensory behavior (*Figure 2M,N*). These results indicate that as ORNs mature, there is a switch in expression from genes important for axon guidance and circuit formation to genes that mediate neuronal signaling and odorant detection. Altogether, our snRNA-seq protocol can robustly and reliably profile developing and adult neurons.

## Individual ORNs maintain expression of the same olfactory receptor(s) throughout development

ORNs use three distinct families of chemoreceptors to detect olfactory stimuli: Ors (odorant receptors), Irs (ionotropic receptors), and Grs (gustatory receptors) (*Clyne et al., 1999*; *Vosshall et al., 1999*; *Scott et al., 2001*; *Benton et al., 2009*). These chemoreceptor genes are gradually turned on in the pupal stage; by the time they reach adulthood, each neuron type expresses one or few specific receptor(s) (*Figure 3A*). Chemosensory receptors can, therefore, serve as excellent ORN type-specific markers. In the mouse olfactory system, an immature ORN can express several olfactory receptors before selecting the one whose expression will be maintained in the mature cell (*Hanchate et al., 2015*; *Tan et al., 2015*). Chemosensory receptor specification has not been thoroughly evaluated in *Drosophila* ORNs during development. Thus, we examined this issue more closely.

Our scRNA-seq data indicated that some ORNs began to express chemosensory receptors as early as 24 hr APF (*Figure 3B*). Focusing on the Or family, we found that by 42 hr APF roughly one-third of ORNs expressed Ors (*Figure 3C,D*). Of the 298 Or-expressing ORNs at 42 hr APF, 84% expressed only one receptor and 16% express two or three (*Figure 3D*). Some ORN types are known to express more than one Or (e.g. Or19a/b and Or65a/b/c; *Dobritsa et al., 2003*; *Couto et al., 2005*; *Fishilevich and Vosshall, 2005*; *Goldman et al., 2005*). Prior to this study *Or85c* was only reported to be expressed in the larval olfactory system (*Kreher et al., 2005*), but we detect co-expression of *Or85c* with *Or85b* in 4% of Or-expressing ORNs (*Figure 3D*). This may be the result of shared regulatory mechanisms controlling the expression of *Or85c* and *Or85b* as these genes are located <1 kb apart, a finding that has been reported for other olfactory receptors separated by a similar genomic distances (*Ray et al., 2007*). All other co-expressed Ors appeared to be randomly paired (*Figure 3D*). Low-frequency co-expression could be accidental or the result of doublet sequencing, an infrequent event whereby the transcriptomes of two cells are inadvertently combined during sample processing or sequencing (*Ilicic et al., 2016*). We conclude that the majority of developing ORNs express only one Or gene.

Do ORNs maintain expression of the same Or from the pupal to adult stage? To address this question, we employed a permanent genetic labeling strategy. We used an *Or-GAL4* to drive *UAS-FLP* to excise the STOP cassette between the ubiquitous *actin5C* promoter and *GAL4* transcriptional activator, such that any cell that transiently expresses the *Or-GAL4* will be labeled. *Or-GAL4* driving expression of *UAS-mCD8-GFP* served as the 'standard labeling' control. We tested *GAL4* lines of three highly expressed Ors at 42 hr APF: *Or42b*, *Or43b*, and *Or47b* using both labeling methods (*Figure 3—figure supplement 1A*). In fewer than 13% of antennal lobes examined, we observed additional labeling outside of the normally innervated glomerulus (*Figure 3E*, *Figure 3—figure*

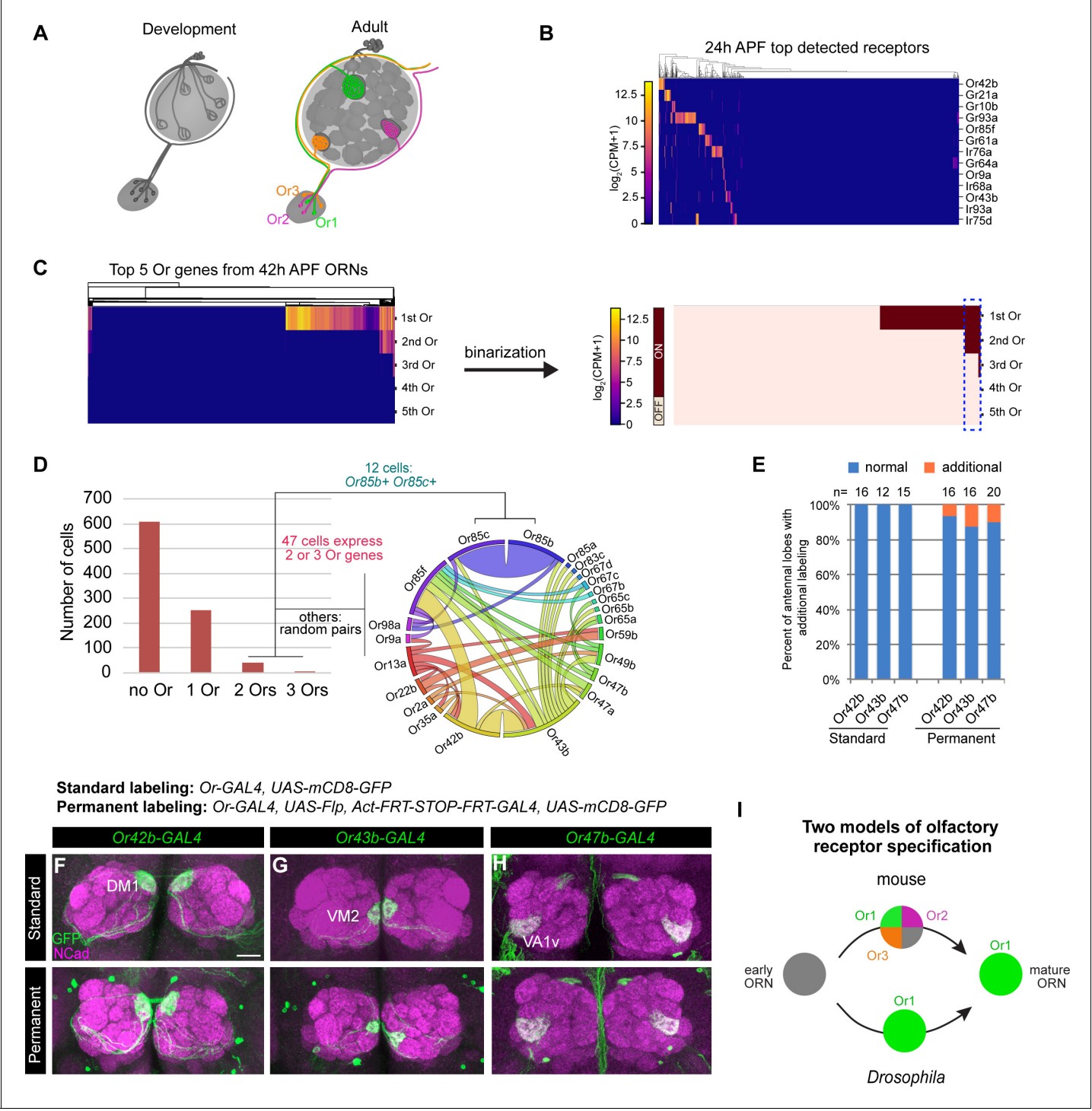

**Figure 3.** Individual ORNs maintain expression of the same olfactory receptor(s) throughout development. (**A**) Schematic depicting ORNs at an early developmental stage (left) and in the mature circuit (right) where each ORN type expresses a distinct olfactory receptor represented by different colors. (**B**) Heatmap of the top detected [$\log_2$(CPM+1) ≥ 3 in > 5 cells] receptor genes in 24 hr APF ORNs. Co-receptors were excluded from this plot. (**C**) Heatmap (left) and binarization of heatmap (right) of the top five highest expressed Ors in 908 *nSybGAL4>mCD8-GFP* positive ORNs from 42 hr APF. Olfactory receptors were considered 'on' if they were expressed at $\log_2$(CPM+1) ≥ 3. Blue dashed box indicates cells that co-express more than one receptor. (**D**) Quantification of the number of cells expressing zero, one, two, or three Ors in 42 hr APF ORNs (left). Chord plot (right) of co-expressed receptors in the cells that expressed 2 or 3 Ors. Most co-expressed receptors show random patterns of expression. (**E**) Quantification of the percentage of antennal lobes that displayed labeling of normal (blue) and additional glomeruli (orange) in standard and permanent labeling conditions. In permanently labeled brains, 6.3% of *Or42b-GAL4;* 12.5% of *Or43b-GAL4;* and 10% of *Or47b-GAL4* antennal lobes displayed additional labeling. (**F–H**)

*Figure 3 continued on next page*

*Figure 3 continued*

Confocal images of adult antennal lobes labeled with anti-GFP (green) and anti-NCad (magenta) in standard labeling (top row) or permanent labeling (bottom row) conditions of *Or42b-GAL4*-positive (F), *Or43b-GAL4*-positive (G), and *Or47b-GAL4*-positive (H) ORNs. Scale bar, 20 μm. (I) Schematic depicting olfactory receptor specification in mice and *Drosophila*. Mouse ORNs express multiple olfactory receptors before one predominates in mature ORNs (top; *Hanchate et al., 2015*; *Tan et al., 2015*), whereas our data indicate that *Drosophila* ORNs maintain expression of the same receptor throughout development (bottom).

The online version of this article includes the following figure supplement(s) for figure 3:

**Figure supplement 1.** Permanent labeling reveals that *Drosophila* ORNs express the same receptor across all developmental stages.

supplement 1B–D'). However, this additional labeling was partially contributed by GAL4 expression in non-ORN neurons innervating the antennal lobe, such as PNs or local interneurons (*Figure 3—figure supplement 1E–G*). Thus, in the vast majority of cases, permanently labeled ORN axons projected to the same glomerulus as the standard labeled controls (*Figure 3E–H*), indicating that these Ors were not expressed in multiple ORN types during development. *Drosophila* and mouse ORNs, therefore, appear to use different strategies for selecting specific olfactory receptor expression (*Figure 3I*).

The finding that olfactory receptor expression is stably maintained across development in *Drosophila* enabled us to use cluster-specific receptor expression to match transcriptomic clusters of developing ORNs to their respective ORN types, which also corresponds to the glomerulus their axons innervate (*Couto et al., 2005*; *Fishilevich and Vosshall, 2005*; *Silbering et al., 2011*).

## Mapping 24 hr APF transcriptomic clusters to their glomerular types

We next investigated ORN transcriptomes at 24 hr APF. When plotted following dimensionality reduction with 500 over-dispersed genes, 24 hr APF transcriptomes were separated into three distinct clusters (*Figure 4—figure supplement 1A*). *acj6*, a transcription factor ubiquitously expressed at 42 hr APF (*Li et al., 2020*), was enriched in the largest cluster but was absent from the other two clusters (*Figure 4—figure supplement 1B*). Antibody staining confirmed that Acj6 protein was expressed in a subset of pan-neuronal Elav+ nuclei in the third antennal segment at 24 hr APF and was absent in the second antennal segment that contains auditory neurons (*Figure 4—figure supplement 1C*). The small *acj6*-negative cluster expressed hearing-related genes (e.g. *iav*; *Senthilan et al., 2012*) and GO terms, suggesting that this cluster corresponds to auditory neurons (*Figure 4—figure supplement 1B,D-F*). The large *acj6*-negative cluster expressed genes and GO terms relating to neurogenesis and sensory organ development, whereas neurite development and cell signaling were found in *acj6*-positive cells (*Figure 4—figure supplement 1F,G*). It is likely that large *acj6*-negative cluster corresponds to ORNs at an early developmental stage, whereas the *acj6*-positive cluster corresponds to more mature ORNs. Thus, the subsequent analyses on 24 hr APF transcriptomes will focus only on the *acj6+* cluster (hereafter 24 hr APF ORNs).

Next, we sought to map 24 hr APF ORNs to their respective glomerular types (*Figure 4A*). In the companion manuscript that analyzed single-cell transcriptomes of PNs (*Xie et al., 2021*), we demonstrated that using differentially expressed genes from the stage with maximal transcriptomic diversity between different cell types for dimensionality reduction achieved the best clustering results. For the three stages of ORNs that we profiled, we found that transcriptomes at 42 hr APF exhibited highest diversity (*Figure 4B*). Thus, we used differentially expressed genes from 42 hr APF ORNs in combination with known ORN type-specific antennal sensory receptors (*Grabe et al., 2016*) as the gene set for dimensionality reduction; this enabled us to obtain clear separation of 24 hr APF transcriptomes when embedded in the UMAP space (*McInnes et al., 2018*; *Figure 4—figure supplement 2A*). This same gene set was used for clustering and dimensionality reduction at all other stages investigated in this study.

We used *meta-learned representations for single cell* (MARS; *Brbić et al., 2020*) to cluster 24 hr APF ORNs. MARS groups cells according to their cell types using a deep neural network to learn a set of cell-type specific landmarks and embedding function to project cells into a latent low-dimensional space. MARS separated 24 hr APF transcriptomes into 36 distinct clusters (*Figure 4—figure supplement 2C*), which corresponded well with clusters identified by the density-based HDBSCAN approach (*Campello et al., 2013*). However, MARS separated closely related clusters that represent

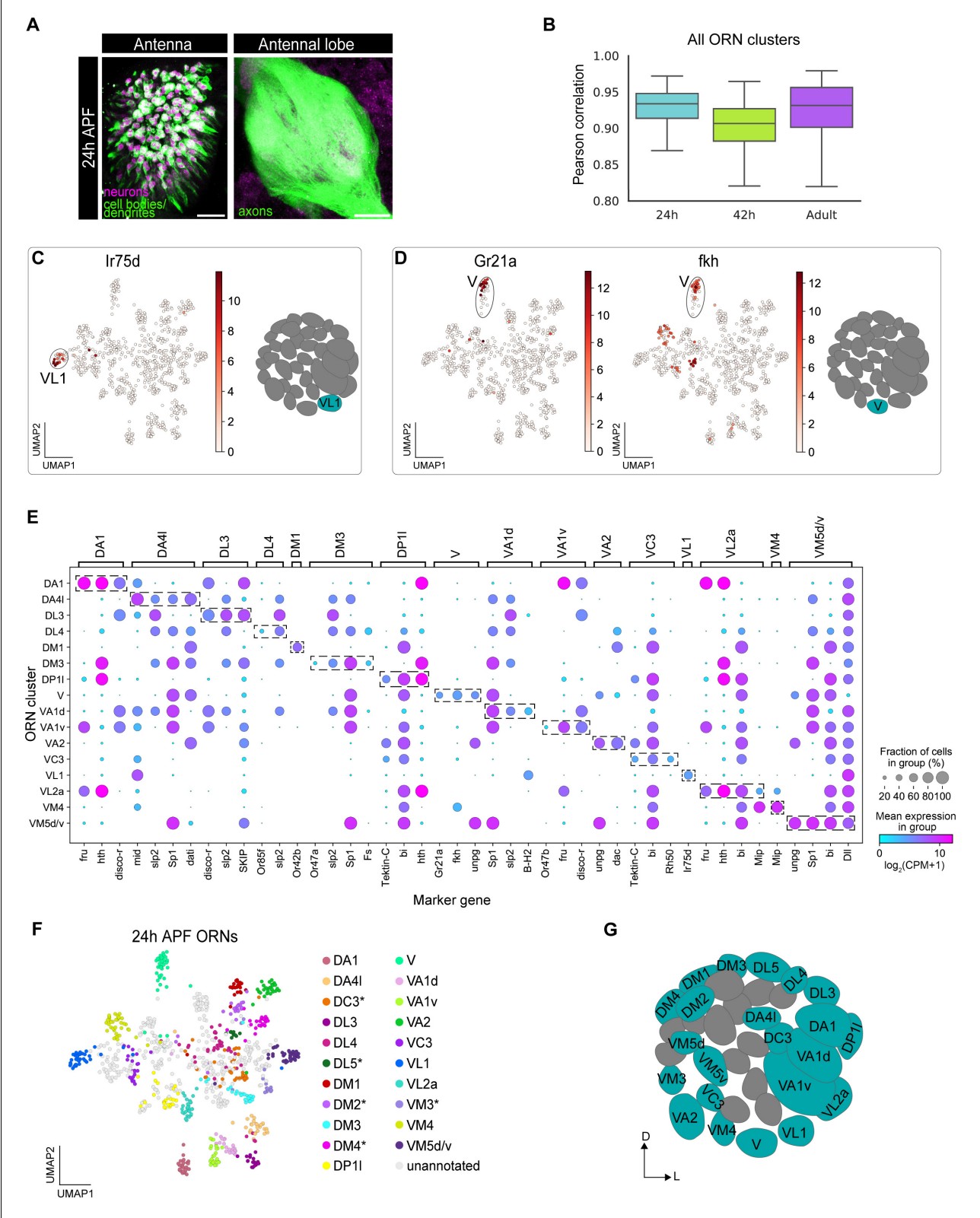

**Figure 4.** Mapping 24 hr APF transcriptomic clusters to their glomerular types. (**A**) Confocal images of 24 hr APF *ey-FLP;UAS-FRT-STOP-FRT-mCD8-GFP;elav-GAL4* ORNs in the antenna (left) labeled with anti-GFP (green) and anti-Elav (magenta) and antennal lobe (right) labeled with anti-GFP (green). Scale bar, 20 μm. (**B**) Quantification of cluster-level transcriptomic similarity of all ORN clusters annotated in this study and in *Li et al., 2020* using Pearson's correlation. 416 differentially expressed genes identified from all three stages were used for this analysis. (**C**) Uniform manifold approximation

*Figure 4 continued on next page*

*Figure 4 continued*

and projection (UMAP) plot depicting *Ir75d* expression in a single cluster (left). *Ir75d*-positive ORNs target their axons to the VL1 glomerulus as shown in the schematic (right). (D) UMAP plots (left) depicting *Gr21a* expression in a single cluster and *fkh* expression in the same cluster. *Gr21a*-positive ORNs target their axons to the V glomerulus and *fkh* is a gene expressed in V ORNs at 24 hr and 42 hr APF. Schematic of the glomerular projection of V ORNs (right). (E) Dot plot summarizing the markers used to map 16 transcriptomic clusters at 24 hr APF to their corresponding 42 hr APF clusters. Dashed boxes around dots highlight markers used to map 24 hr APF clusters to 42 hr APF ones. (F) UMAP plot depicting the 21 transcriptomic clusters that were decoded at 24 hr APF. Asterisk next to ORN type indicates that it was annotated using data presented in **Figure 6**. Note that VM5d/v could not be separated (see **Figure 4—figure supplement 2C**). (G) Schematic depicting the glomerular targeting patterns of the 22 ORN types mapped in E–F and **Figure 6**. Heatmap units, log$_2$(CPM+1). For all UMAP plots: dimensionality reduction was performed using differentially expressed genes from 42 hr APF combined with sensory receptors for a total of 335 genes.

The online version of this article includes the following figure supplement(s) for figure 4:

**Figure supplement 1.** Further analysis of 24 hr APF transcriptomic clusters.

**Figure supplement 2.** Quality control and additional analysis of 24 hr APF transcriptomes of Acj6-positive ORNs.

**Figure supplement 3.** Markers used to match 24 hr APF ORNs to their glomerular type.

multiple ORN types better than HDBSCAN (**Figure 4—figure supplement 2C,D**). In fact, MARS was able to separate two clusters, VM5d and VM5v (**Figure 4—figure supplement 2E**), in our 42 hr APF data that were previously indistinguishable (**Li et al., 2020**). Neither sequencing batch nor sequencing depth drove clustering (**Figure 4—figure supplement 2B**).

Having clustered neurons based on their transcriptomic differences, we first asked if we could detect cluster-specific expression of any sensory receptors at 24 hr APF, as receptor expression in a single cluster would enable us to immediately identify the glomerular type the cluster represents. Cluster-specific expression of six receptors allowed us to annotate clusters corresponding to V (*Gr21a*-expressing), DM1 (*Or42b*), VL1 (*Ir75d*), DL4 (*Or85f*), VA1v (*Or47b*), and DM3 (*Or47a*) ORN types (**Figure 4C–E**, **Figure 4—figure supplement 3A–D**).

We next used marker genes to match additional 24 hr APF clusters with their corresponding 42 hr APF clusters. We first asked if cluster-specific markers from 42 hr APF displayed comparable expression at 24 hr APF and could be used to annotate clusters at this earlier stage. Demonstrating the feasibility of this strategy, *forkhead* (*fkh*) was expressed in neurons projecting to the V glomerulus at both 24 hr and 42 hr APF (**Figure 4D** middle, E, **Figure 4—figure supplement 2E**; **Li et al., 2020**). We identified a set of additional marker genes whose individual or combinatorial expression was sufficient to map ORNs at 24 hr APF to their respective 42 hr APF clusters (**Figure 4—figure supplement 2E**). Using this approach, we decoded the glomerular identity of an additional 10 ORN clusters (**Figure 4E–G**, **Figure 4—figure supplement 3**). Two of these ORN types, VM5d and VM5v, mapped to a single cluster at 24 hr APF (**Figure 4E**), indicating that these cells are highly similar at this stage. Collectively, we mapped 16 transcriptomic clusters to 17 ORN types at 24 hr APF. Five additional clusters were mapped to ORN types by matching transcriptomic clusters across development (see Figure 6), bringing total mapped 24 hr APF ORN clusters to 21.

## Mapping adult transcriptomic clusters to their glomerular types

The adult antenna is covered in hair-like projections called olfactory sensilla (**Figure 5A**). Each sensillum comprises 1–4 ORNs and accessory cells that are all derived from a common progenitor (**Figure 5A′**; **Endo et al., 2012**). Antennal ORNs are located in one of four morphologically distinct sensillar classes: trichoids, coeloconics, large basiconics, and thin/small basiconics (**Figure 5B**). Thermo- and hygrosensory neurons are found within the sacculus, a three chambered invagination within the antenna, and the arista (**Figure 5B**; **Enjin et al., 2016**; **Budelli et al., 2019**). All adult ORNs should express their type-specific receptor(s), enabling us to definitively map adult clusters to their glomerular type. We considered a sensory receptor to be expressed if it was present in more than five neurons at log$_2$(CPM+1) $\geq$ 3. Using this criterion, 84% of previously reported antennal sensory receptors (**Menuz et al., 2014**) were expressed in adult transcriptomes (**Figure 5C**). Excluding widely expressed co-receptors (see Figure 9), more than 80% of adult neurons express a sensory receptor (**Figure 5C**). Dropouts and low-level receptor expression likely account for the less than 20% of neurons where we could not detect a receptor, as these receptors exhibit a large dynamic range of expression (**Menuz et al., 2014**). We also compared the sensory receptors detected using snRNA-seq with those observed by bulk antennal RNA-seq (**Menuz et al., 2014**; **Barish et al., 2017**)

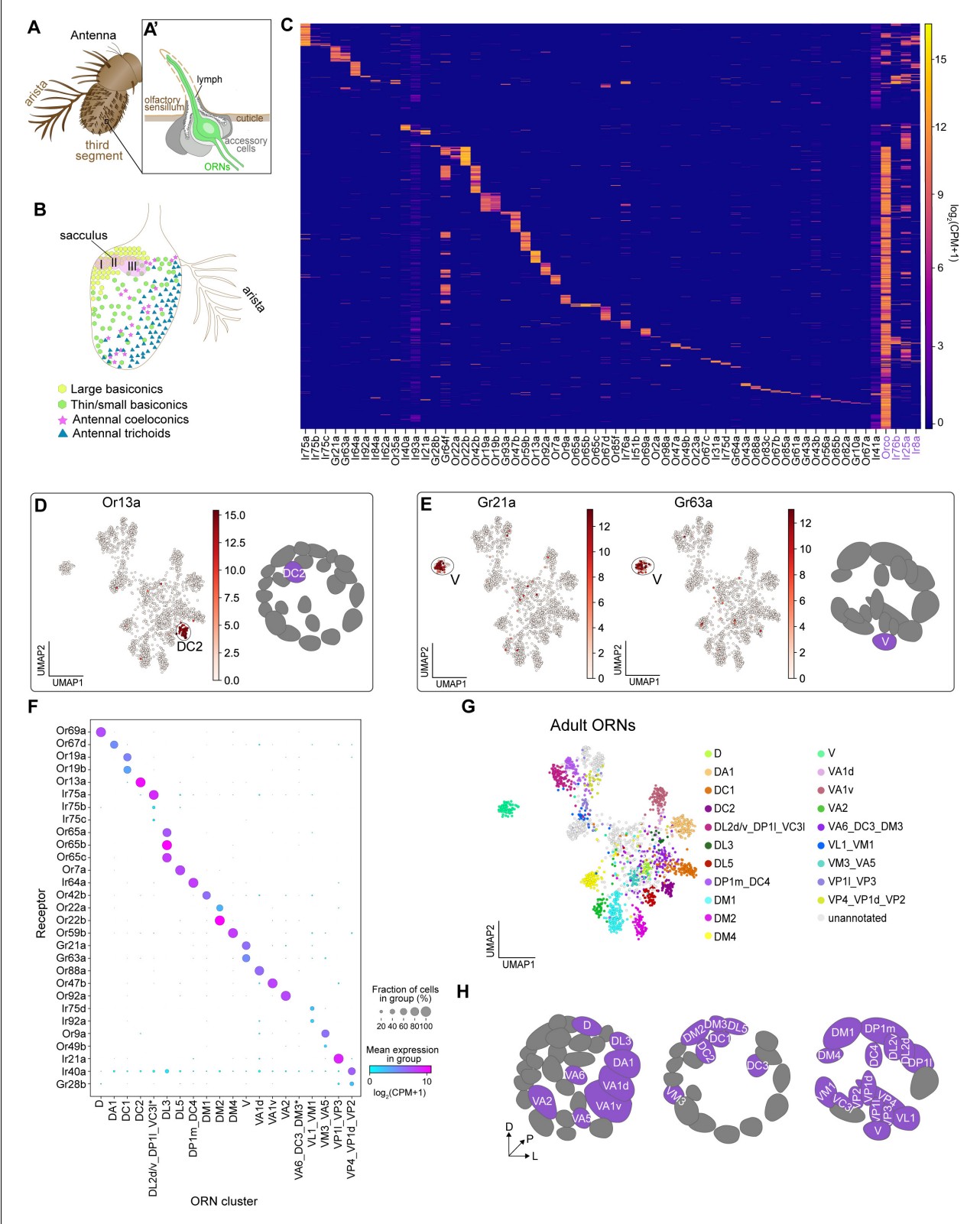

**Figure 5.** Mapping adult transcriptomic clusters to their glomerular types. (**A–A'**) Diagram of the adult antenna (**A**) and the structure of an individual sensillum (**A'**) where ORN dendrites and support cells are located. (**B**) Schematic of the antenna depicting the spatial distribution of sensillar classes and the location of the sacculus and arista. (**C**) Heatmap showing the expression of all detected sensory receptors belonging to the Or (odorant receptor), Gr (gustatory receptor), and Ir (ionotropic receptor) families. A detected receptor is expressed at a level of log₂(CPM+1) ≥3 in > 5 cells. Note low level

*Figure 5 continued on next page*

*Figure 5 continued*

expression of *Ir41a* and *Ir93a* in many cells may be the result of increased read numbers generated by their large genomic regions. Or and Ir co-receptors are in purple. Cells were ordered by hierarchical clustering. (D–E) UMAP plots depicting *Or13a* expression in a single cluster (left) and glomerular projection of *Or13a*-positive DC2 ORNs (right) (D) and *Gr21a* and *Gr63a* expression in a single cluster (left) and glomerular projection of *Gr21a/63a*-positive V ORNs (E). (F) Dot plot depicting sensory receptors used to annotate adult neuron clusters. Clusters that contained multiple neuron types are indicated by an underscore separating the names of each type. Asterisk next to VA6_DC3_DM3 and DP1l_DL2d/v_VC3l clusters indicates that some or all of receptors for these cell types are expressed in a small non-overlapping percentage of cells and can be better resolved on a UMAP plot (see *Figure 5—figure supplement 2K, M*). (G) UMAP plot of 20 adult clusters annotated to 31 neuron types using sensory receptor expression. Note that some clusters contained receptors of multiple ORN types and are labeled as such. (H) Schematic of the glomerular targets of the 31 neuron types mapped in F and G. For all UMAP plots, dimensionality reduction was performed using differentially expressed genes from 42 hr APF combined with sensory receptors for a total of 335 genes. Heatmap units, $\log_2(CPM + 1)$.

The online version of this article includes the following figure supplement(s) for figure 5:

**Figure supplement 1.** Quality control and additional analysis of adult transcriptomes.

**Figure supplement 2.** Cluster-specific sensory receptor expression in adult transcriptomes.

and found that these two technologies detect largely overlapping receptors (*Figure 5—figure supplement 1A*).

Using similar methods as described earlier, we found that adult ORN transcriptomes were divided into 26 clusters (*Figure 5—figure supplement 1D*). Similar to 24 hr APF, MARS outperformed density-based clustering of adult ORNs (*Figure 5—figure supplement 1D,E*). Although our adult transcriptomes included nuclei labeled with both *unc84-GFP* and *lam-GFP*, neither batch effects nor sequencing depth drove clustering (*Figure 5—figure supplement 1F,G*).

Do adult transcriptomic clusters represent distinct neuron types? We observed cluster-specific expression of many Ors and Grs in our data (*Figure 5D–E*, *Figure 5—figure supplement 2*). For instance, *Or13a* (expressed by DC2-projecting ORNs) was found in a single cluster (*Figure 5D*). Multiple other Ors also exhibited expression in a single cluster (*Figure 5F*, *Figure 5—figure supplement 2*), which enabled us to determine the ORN type identity of these clusters. Further, we observed cluster-specific expression of all of the known co-expressed receptors for the following ORN types: V (*Gr21a/Gr63a*), DL3 (*Or65a/b/c*), DC1 (*Or19a/b*), and DM2 (*Or22a/b*) (*Figure 5E,F*, *Figure 5—figure supplement 2B,C,J*).

In some instances, we were unable to perform one-to-one annotations of adult clusters to sensory neuron types. This occurred for two reasons. First, sensory receptors from multiple neuron types were intermingled in a single cluster. For instance, both *Or49b* (VA5-projecting ORNs) and *Or9a* (VM3-projecting ORNs) were found in separate populations of cells within the same cluster (*Figure 5F*; *Figure 5—figure supplement 2A*). Thus, we annotated this cluster as containing both VM3 and VA5 ORNs. Second, a subset of Irs are expressed in multiple sensory neuron types that project to distinct glomeruli in the antennal lobe (*Grabe et al., 2016*; *Marin et al., 2020*). As an example, *Ir21a* is expressed in neurons projecting from the antenna and arista to the VP1l and VP3 glomeruli, respectively (*Marin et al., 2020*). Because we could not further sub-divide these *Ir21a*-positive cells, we annotated this cluster as representing VP1l- and VP3-projecting neurons (*Figure 5F*, *Figure 5—figure supplement 2P*). Accordingly, we annotated the following neuron types as being found in the same cluster: (1) DM3 (*Or47a*), VA6 (*Or82a*), and DC3 (*Or83c*); (2) VM1 (*Ir92a*) and VL1 (*Ir75d*); (3) DP1l (*Ir75a*), DL2d/v (*Ir75a/b/c*), and VC3l (*Or35a*); (4) DP1m (*Ir64a*) and DC4 (*Ir64a*); and (5) VP4 (*Ir40a*), VP1d (*Ir40a*), and VP2 (*Gr28b*) (*Figure 5F*, *Figure 5—figure supplement 2*). In most cases, when we observed multiple ORN types (expressing distinct receptors) in the same cluster, the intermingled ORNs originated from either the same sensillar class (e.g. DL2d/v, DP1l, and VC3l) or the same sensillum (e.g. VM1 and VL1), indicating that the transcriptomes of these cells are very similar at the adult stage. This likely contributed to why we have 26 adult clusters instead of ~50 (see Discussion). Collectively, we used cluster-specific receptor expression to map 31 neuron types to 20 adult transcriptomic clusters (*Figure 5G–H*).

## Matching transcriptomic clusters across all stages enabled annotation of additional ORN types

In addition to the annotated ORN types from our prior 42 hr APF study (*Li et al., 2020*), we mapped 17 and 31 ORN types to their corresponding 24 hr APF and adult transcriptomic clusters,

respectively (*Figures 4F*, *5G*). When we plotted the transcriptomes of all stages together as previously described (*Figure 4*), they readily separated into clusters corresponding to distinct ORN types and stages (*Figure 6A,B*). Denoting a high degree of transcriptomic similarity, cells belonging to the same ORN type at 24 hr and 42 hr APF predominantly clustered together, whereas adult transcriptomes remained distinct (*Figure 6A,B*). Differences in either sequencing technology or stage-specific transcriptomic differences could both contribute to why adult transcriptomes did not intermingle with pupal ones.

We next wanted to leverage the similarity between 24 hr and 42 hr ORNs, as well as the fact that we have annotated different neuron types between pupal and adult stages, to decode additional clusters at each time point. Since we already employed a manual matching strategy (using marker genes/sensory receptor expression), we asked if an automatic matching approach that uses transcriptomic similarity to link clusters across development could annotate more clusters (*Figure 6C*). Thus, we identified differentially expressed genes in each cluster at all stages profiled. We then independently compared the transcriptomes of each cluster at one stage to every cluster at the other time points. A similarity score (Jaccard index) was then computed for the proportion of genes shared between each pair of clusters across two stages. If two clusters both had the highest similarity score with each other, we considered these clusters to be two-way matched (*Figure 6C*). One-way matches occurred when one cluster had the highest similarity with a cluster at a different stage, but the opposite did not hold true (*Figure 6C*, see Materials and methods). Due to the difference in sequencing technologies between pupal and adult ORNs, we did not allow one-way matches between these stages.

We first used this approach to validate our manual matches and found that the results of these two strategies were in complete agreement (*Figure 6D*). Our automatic two-way matching approach linked six more clusters between 24 and 42 hr APF. Two of these clusters corresponded to DM4 and VM3 ORNs and four were unannotated (*Figure 6B,D*, *Figure 6—figure supplement 1A,C*). The pupal DL5 clusters were annotated by matching them to the previously annotated adult DL5 cluster (*Figure 6D*). Lastly, we used one-way automatic matching to annotate DC3 and DM2 ORNs at 24 hr APF (*Figure 6D*). Automatic matching was able to decode an additional five transcriptomic clusters at 24 hr APF and three additional clusters at 42 hr APF. Altogether, by combining manual and automatic approaches, we matched 17 ORN clusters across all three developmental stages.

## ORN transcriptomes in part reflect axon trajectory at 24 hr APF and sensory modality in adults

Generally speaking, antennal sensory neurons that are found in the same structure (e.g. sensillar class, arista, or sacculus) detect similar stimuli and project their axons to adjacent regions in the antennal lobe (*Couto et al., 2005*; *Fishilevich and Vosshall, 2005*; *Silbering et al., 2011*). Since we annotated neurons in each of these sensory structures (*Figure 7A*), we next sought to determine the stage-specific transcriptomic features of neurons in each of these groups. Specifically, we probed the relationship between transcriptome, sensillar class, and stage-related biological processes of neurons at each time point.

We began with 24 hr APF transcriptomes. We identified differentially expressed genes between all annotated ORN types (that mapped to a single cluster) and performed hierarchical clustering of their transcriptomes based on expression of these genes. Overall, ORNs that belong to the same sensillar class share more similar transcriptomes than those from other classes (*Figure 7B*), similar to our previous observations at 42 hr APF (*Li et al., 2020*). In some cases (e.g. VA1v and VA1d), the transcriptomes of ORNs within the same sensillum were most similar to each other (*Figure 7B*). Yet, neither V nor DL4 neurons clustered with their sensillar classes. We considered other factors that could contribute to the clustering of these two ORN types. At 24 hr APF, most ORN axons are taking either a dorsolateral or ventromedial trajectory around the antennal lobe (*Endo et al., 2007*; *Joo et al., 2013*). Could axon trajectory explain the clustering of DL4 and V transcriptomes? Indeed, DL4 ORNs were more similar to the antennal trichoids that take a dorsolateral trajectory than to the ventromedially-projecting basiconics annotated here (*Figure 7B*). Both V neurons and the coeloconics they clustered with project their axons to more posterior glomeruli than the majority of ORNs in trichoid and basiconic classes (*Couto et al., 2005*; *Rybak et al., 2016*; *Figure 7B*). Transcriptomes of posteriorly projecting ORNs further segregated based on axon trajectory, as ORNs whose axons take the ventromedial path were more similar to each other than to those taking the dorsolateral

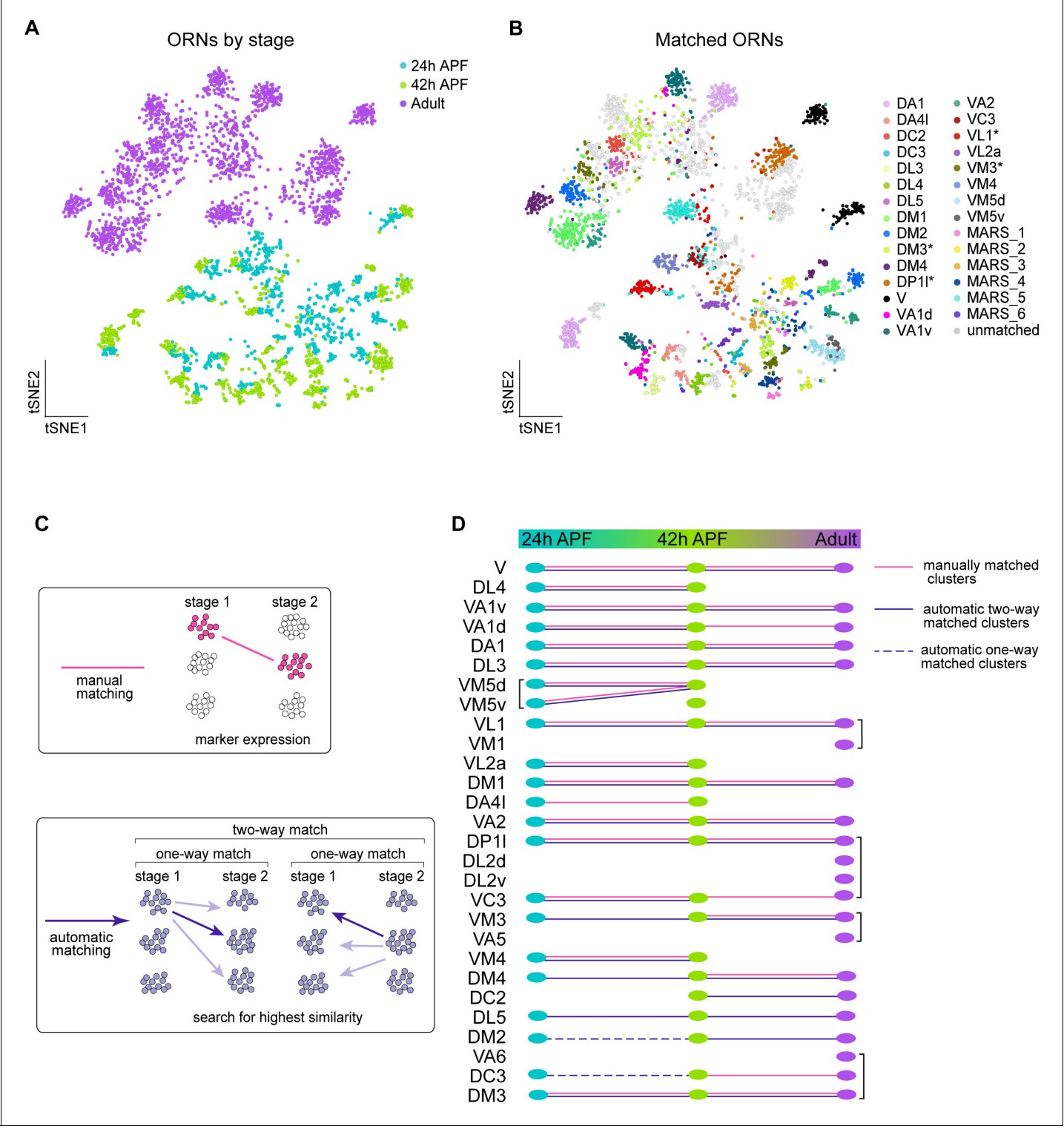

**Figure 6.** Matching transcriptomic clusters across all stages enabled annotation of additional ORN types. (A–B) t-SNE plots of ORNs from 24 hr APF, 42 hr APF, and adult stages (A) and matched ORNs across these stages (B). Clusters marked 'MARS_#' are matched but their glomerular identity is unknown. Asterisk next to cluster name indicates that at the adult stage the cluster contained multiple ORN types, but it was annotated with the name of the developmental cluster it mapped to. Dimensionality reduction was performed using differentially expressed genes from 42 hr APF combined with olfactory receptors for a total of 335 genes. (C) Schematic summarizing the two approaches used for matching ORNs from different stages: (1) manual matching using cluster-specific markers/olfactory receptors (top); and (2) automatic matching by calculating the transcriptome similarity between one

*Figure 6 continued*

cluster at a certain stage to all clusters from another stage (bottom). (D) Summary of all matched ORN types from three different stages, the method used to match clusters is represented by the type of line (color, solid, dashed) linking the clusters between timepoints.
The online version of this article includes the following figure supplement(s) for figure 6:

**Figure supplement 1.** Additional evidence for cluster matching across developmental stages.

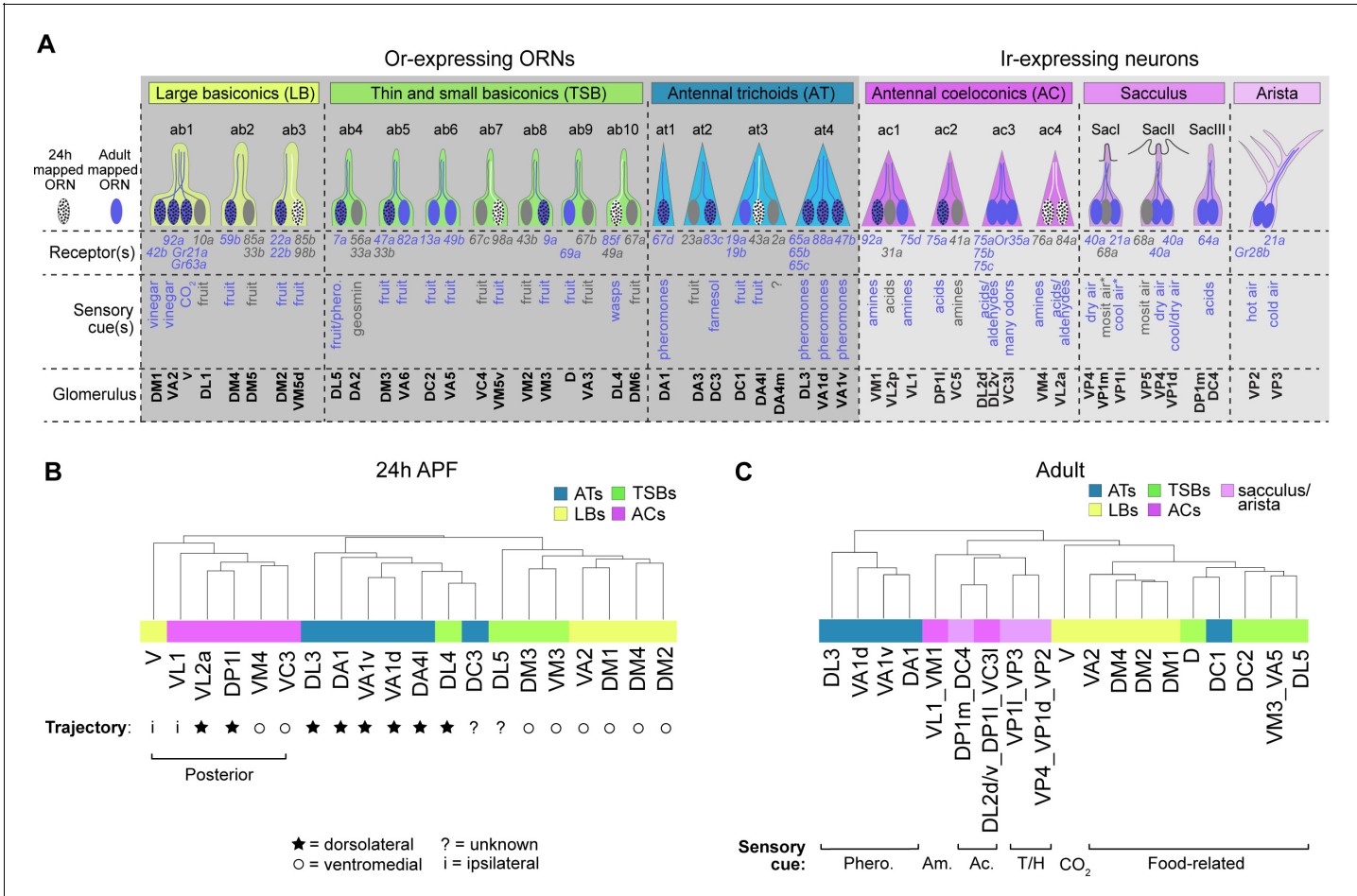

**Figure 7.** ORN transcriptomes in part reflect axon trajectory at 24 hr APF and sensory modality in adults. (A) Schematic summary of ORNs, thermo- and hygro-sensory neurons within their respective sensillar class/structure, the sensory receptors each neuron type expresses, and the category of cues each type predominantly responds to. White dotted ORNs were annotated at 24 hr APF, blue ORNs were annotated in adults, and blue dotted ORNs are annotated at both stages. Receptors in blue were detected in either 24 hr APF or adult datasets. Sensory cues marked with asterisks indicate that they are putative stimuli for that neuron type. See references below for information regarding the correspondence between receptor expression and glomerular projection (*Couto et al., 2005*; *Fishilevich and Vosshall, 2005*; *Silbering et al., 2011*; *Grabe et al., 2016*; *Marin et al., 2020*) and regarding sensory stimuli detected by specific types (*Yao et al., 2005*; *Hallem and Carlson, 2006*; *Kurtovic et al., 2007*; *Kwon et al., 2007*; *Ray et al., 2007*; *Semmelhack and Wang, 2009*; *Gallio et al., 2011*; *Silbering et al., 2011*; *Ni et al., 2013*; *Ronderos et al., 2014 Ebrahim et al., 2015*; *Lin et al., 2015* ; *Enjin et al., 2016*; *Dweck et al., 2013*; *Dweck et al., 2018*; *Budelli et al., 2019*). (B–C) Dendrograms of 24 hr APF (B) and adult (C) sensory neuron types that resulted from hierarchical clustering of transcriptomes based on differentially expressed genes at each stage. Axon trajectory/projection information is from *Endo et al., 2007*; *Silbering et al., 2011*; *Joo et al., 2013*; *Rybak et al., 2016*. The following abbreviations were used to describe the stimuli adult neurons respond to: Phero. = pheromones; Am = amines; Ac = acids; and T/H. = temperature and humidity. Some adult clusters contain multiple neuron types, but these clusters are within the same sensillar class or detect the same cues.
The online version of this article includes the following figure supplement(s) for figure 7:

**Figure supplement 1.** Comparisons of 24 hr APF ORNs by sensillar class.
**Figure supplement 2.** Comparisons of adult ORNs by sensillar class and sensory structure.

trajectory. Further, V and VL1 ORNs have the most divergent transcriptomes of the posteriorly projecting ORNs (*Figure 7B*), which could reflect the fact that these neurons send their axons ipsilaterally to one antennal lobe and not bilaterally to both lobes, like all the other ORNs in the dendrogram (*Couto et al., 2005*; *Silbering et al., 2011*). We also note that cell-surface molecules and transcription factors, which are important regulators of axon targeting, constituted 14% and 25% (as opposed to less than 6% each in the genome; *Figure 8B*) of the differentially expressed genes used to cluster ORN transcriptomes, respectively (*Figure 7—figure supplement 1A–B*). These data suggest that, in addition to sensillar class, similarities in 24 hr APF ORN transcriptomes reflect similarities in axon trajectory.

Our adult data include annotated ORNs as well as thermosensory and hygrosensory neurons, which enabled us to assess the relationship between multiple sensory neuron types at the transcriptome-level. Similar to the pupal stages, hierarchical clustering with differentially expressed genes indicated that neurons within the same sensory structures shared more similar transcriptomes (*Figure 7C*, *Figure 7—figure supplement 2*). A closer examination revealed that adult transcriptomes also reflected the type of stimuli they detect. DC1 ORNs from the trichoid at3 sensillum, for instance, sense limonene (*Dweck et al., 2013*) and their transcriptomes were more similar to fruit odor-responsive basiconic ORNs than to the pheromone-detecting trichoid ORNs (*Figure 7A,C*). In fact, DC1 transcriptomes were most similar to D ORNs which can also respond to limonene (*Dweck et al., 2013*). Within the Ir-expressing clade, neurons also appeared to group by the stimuli they respond to. For example, acid-sensing neurons in the coeloconic sensilla were more closely related to sacculus neurons detecting the same type of stimulus than to amine-sensing coeloconics (*Figure 7A,C*). Further indicating transcriptomic similarity also reflects sensory modality, the thermo- and hygrosensory neurons in the sacculus and arista were most similar to each other (*Figure 7C*) and in some cases inseparable at the cluster level (*Figure 5G*). Taken together, our data suggest that ORN transcriptomes in part reflect stage-related biological processes.

## Cell-type-specific expression of transcription factors and cell-surface molecules in developing ORNs

We next investigated the genes that may contribute ORN wiring in 24 hr APF and 42 hr APF transcriptomes, as these stages represent axon trajectory choice and synaptic partner matching, respectively. We focused on uncovering genes that are differentially expressed between ORNs at these two stages, especially in the 21 ORN types whose transcriptomes were matched between 24 and 42 hr APF (*Figure 6*).

We first calculated differentially expressed genes (adjusted p-value < 0.01 by Mann-Whitney U test) in at least one cluster compared to the rest of the clusters at each stage. We identified 370 and 554 differentially expressed genes between all 24 hr APF and 42 hr APF ORN clusters, respectively (*Figure 8A*). The fact that 42 hr APF ORNs had more differentially expressed genes than their 24 hr APF counterparts is in line with our finding that this mid-developmental stage contained ORNs with the most distinct transcriptomes (*Figure 4B*). Next, we defined the set of differentially expressed genes shared between both developmental stages, as this dataset likely contains genes important for ORN development. This list comprised 211 genes, 170 of which were differentially expressed among the 21 matched ORN types (*Figure 8A*). Transcription factors (TFs) and cell-surface molecules (CSMs) were over-represented in the differentially expressed genes shared between 24 hr and 42 hr APF ORNs compared to their genome-wide percentages (*Figure 8B*; *Figure 8—figure supplements 1–4*). TFs and CSMs are important regulators of ORN development and wiring (*Hong and Luo, 2014*; *Li et al., 2020*), so we further evaluated their expression in matched developing ORNs.

Thirty-four TFs were differentially expressed in the developing ORNs matched between 24 and 42 hr APF. Some TFs displayed broad expression in nearly all ORN types, including *tonali* (*tna*), *scribbler* (*sbb*), *Distal-less* (*Dll*), *48 related 1* (*Fer1*), and *Ecdysone-induced protein 93F* (*Eip93F*) (*Figure 8C*), the latter four of which have previously described expression in the antenna (*Jafari et al., 2012*; *Barish et al., 2017*; *Chai et al., 2019*). Both Fer1 and Eip93F regulate expression of a subset of Or genes (*Jafari et al., 2012*), but our data suggests that they could have additional functions in Ir-positive ORNs (*Figure 8C*). These analyses also revealed TFs that were expressed in specific sensillar classes, including *extra macrochaetae* (*emc*) in coeloconic and V ORNs (both of which project their axons to posterior glomeruli), *datilografo* (*dati*) is largely in basiconic ORNs, and *eyegone* (*eyg*) and *twin of eyg* (*toe*) in trichoid and thin/small basiconic ORNs

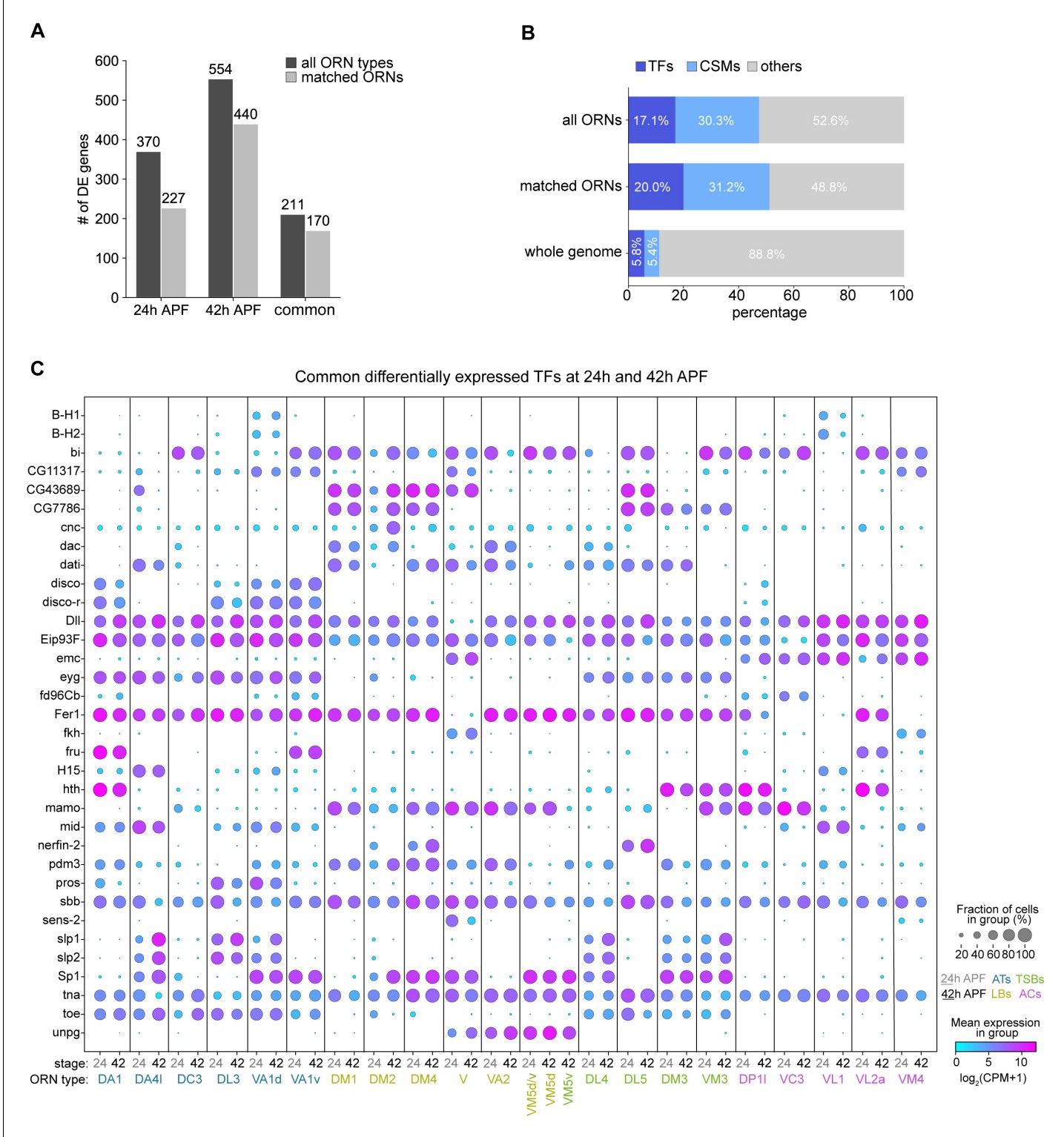

**Figure 8.** Cell-type-specific expression of transcription factors (TFs) and cell-surface molecules (CSMs) in developing ORNs. (A) Quantification of differentially expressed (DE) genes identified in all ORN clusters (dark gray) or in clusters matched between 24 and 42 hr APF (light gray) in each pupal stage. (B) Percentage of TFs or CSMs from the common list of DE genes from 24 and 42 hr APF ORNs compared to the genome-wide percentage. (C) Dot plot depicting expression of the 34 common DE TFs from both 24 hr APF and 42 hr APF stages.

The online version of this article includes the following figure supplement(s) for figure 8:

**Figure supplement 1.** Common differentially expressed CSMs in 24 hr and 42 hr APF ORNs.

*Figure 8 continued on next page*

**Figure supplement 2.** All differentially expressed CSMs in 24 hr and 42 hr APF ORNs, Part I.
**Figure supplement 3.** All differentially expressed CSMs in 24 hr and 42 hr APF ORNs, Part II.
**Figure supplement 4.** All differentially expressed TFs in 24 hr or 42 hr APF ORNs.
**Figure supplement 5.** CSM expression in synaptic partner ORNs and PNs, Part I.
**Figure supplement 6.** CSM expression in synaptic partner ORNs and PNs, Part II.

(*Figure 8C*). Finally, some TFs showed restricted expression to a few ORN types. For instance, *nervous fingers 2* (*nerfin-2*) was expressed by DM4 and DL5 ORNs at both stages and DM2 ORNs at 24 hr APF, and *senseless-2* (*sens-2*) was expressed by V and, less strongly, by VM4 ORNs (*Figure 8C*). *BarH1/BarH2* (*B-H1/B-H2*) were expressed in VA1d and VL1 ORNs (*Figure 8C*), in line with a previous report (*Li et al., 2016*). *Fruitless* (*Fru*) started to be differentially expressed at 24 hr APF in the same three ORN types as previously reported in adults (*Stockinger et al., 2005*). Supporting the notion that ORN types are defined by a combinatorial code of TFs, expression of a single TF was insufficient to distinguish any ORN types; instead this required expression of multiple TFs.

We next analyzed CSMs. Fifty-three CSMs were differentially expressed in developing ORNs (*Figure 8—figure supplement 1*). Many of them were members of protein families implicated in circuit assembly, such as Beaten Path (beat), Down Syndrome Cell Adhesion Molecule (Dscam), Fasciclin (Fas), Netrin (Net), Roundabout (robo), Defective Proboscis Extension Response (Dpr), and Semaphorin (Sema; *Figure 8—figure supplements 1–3*; *Kolodkin and Tessier-Lavigne, 2011*; *Sanes and Zipursky, 2020*), suggesting that these differentially expressed CSMs could regulate ORN wiring. Similar to other scRNA-seq studies of developing neurons (*Kurmangaliyev et al., 2020*; *Özel et al., 2021*; *Xie et al., 2021*), many CSMs exhibited graded and/or broad expression in multiple neuron types. *Semaphorin 2B* (*Sema-2b*) and *Teneurin-a* (*Ten-a*), which display restricted expression at the protein level (*Hong et al., 2012*; *Joo et al., 2013*), were broadly expressed at the mRNA level (*Figure 8—figure supplement 1*), indicating post-transcriptional regulation of some of these CSMs. Expression of a subset CSMs was restricted to distinct sensillar classes: *Wnt oncogene analog 4* (*Wnt4*) was expressed by many trichoid and coeloconic ORNs; *Connectin* (*Con*) was enriched in trichoid and basiconic classes; and *Fasciclin* (*Fas3*) was highly expressed by coeloconic ORNs (*Figure 8—figure supplement 1*). Lastly, a few CSMs, such as *beaten path IIa and IIIc* (*beat-IIa* and *beat-IIIc*), *Netrin-B* (*NetB*), and *no long nerve cord* (*nolo*), showed limited expression in a few ORN types (*Figure 8—figure supplement 1*).

Finally, to aid in identification of molecules important for synaptic partner matching, we queried differentially expressed genes in six synaptic partner ORNs and PNs (from the companion manuscript, *Xie et al., 2021*) at 42 hr APF and 48 hr APF, respectively. From this gene set, we identified 203 differentially expressed CSMs (*Figure 8—figure supplements 5*, *6*). Some of these CSMs were broadly expressed in nearly all ORN and PN pairs (e.g. *Fasciclin1* [*Fas1*] and *Basigin* [*Bsg*]), whereas other genes showed enrichment in either ORNs or PNs (e.g. *Neuroligin2* [*Nlg2*] in PNs and *bride of sevenless* [*boss*] in ORNs). Identifying ligand-receptor pairs (e.g. *Özkan et al., 2013*) and determining whether their interactions lead to attraction or repulsion will be crucial to decipher how CSMs expressed in ORNs and PNs mediate synaptic partner matching. Taken together, we have identified multiple differentially expressed CSMs and TFs in distinct developing ORN types, providing a useful resource for future studies of ORN development and wiring specificity.

## Broad co-receptor expression in adult ORNs

Obtaining adult single-cell transcriptomes enabled us to explore gene expression differences between distinct types of antennal sensory neurons. We began by classifying these neurons based on the two major classes of sensory receptors they express: Ors and Irs. Ors are used to detect odorants, whereas Irs can respond to odorants, temperature, and humidity (*Hallem and Carlson, 2006*; *Silbering et al., 2011*; *Knecht et al., 2016*; *Knecht et al., 2017*; *Ni et al., 2016*). Despite having seven transmembrane domains, insect Ors are not G-protein-coupled receptors (GPCRs) but instead adopt an inverse seven-transmembrane topology with an intracellular amino-terminal domain and an extracellular carboxy-terminal domain (*Figure 9A*; *Benton et al., 2006*). Ors assemble into heterotetramers with their obligate co-receptor, Orco, to form ligand-gated cation channels

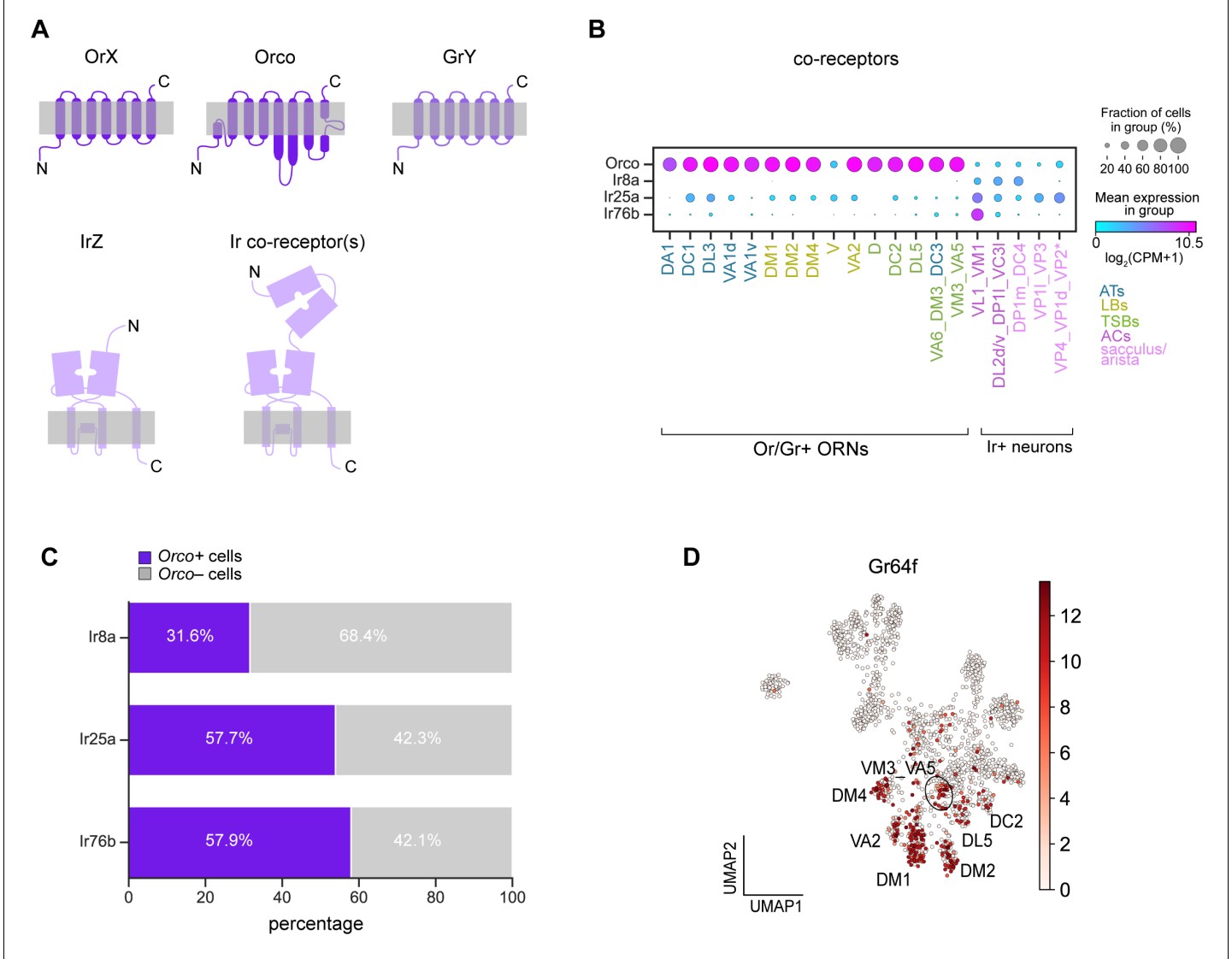

**Figure 9.** Broad co-receptor expression in adult ORNs. (A) Schematic of three classes of *Drosophila* sensory receptors. OrX (odorant receptors) and their co-receptor Orco are seven-pass transmembrane receptors, as are GrY (gustatory receptors) (top). IrZ (ionotropic receptors) and their co-receptors are relatives of ionotropic glutamate receptors. Orco diagram is based on *Butterwick et al., 2018* and Ir/Ir co-receptor diagram is based on *Abuin et al., 2019*. (B) Dot plot depicting Or and Ir co-receptor expression in adult clusters. Clusters containing multiple neuron types are noted using an underscore to separate names of each type. Asterisk next to VP2 ORNs indicates that these cells express a Gr and not an Ir. We observe *Orco* co-expression in many of the same ORN types as *Task et al., 2020*; however, although *Orco* expression is substantially lower than that of *Obps*, we could not fully rule out the possibility that *Orco* expression in a subset of cells was due to ambient contamination as was likely the case of *Obps* (*Figure 2—figure supplement 1*). (C) Stacked bar graph of the percentage of cells that express each Ir co-receptor that also express *Orco*, with a stringent expression cut-off of $\log_2(\text{CPM}+1) \geq 5$ for all co-receptors (based on *Figure 9—figure supplement 1F–H*). (D) UMAP plot of *Gr64f* expression in adult ORNs.

The online version of this article includes the following figure supplement(s) for figure 9:

**Figure supplement 1.** Expression of *Orco* and Ir co-receptors in adult ORNs.

**Figure supplement 2.** Expression of some gustatory receptors is correlated with the expression of a subset of olfactory receptors.

(*Butterwick et al., 2018*). Irs are glutamate-insensitive relatives of ionotropic glutamate receptors (iGluRs) and are proposed to exist as heteromeric cation channels with their co-receptors (*Figure 9A*; *Abuin et al., 2019*). V- and VP2- projecting neurons do not use Ors or Irs to detect stimuli; they instead use Grs to sense $CO_2$ and heat, respectively (*Kwon et al., 2007*; *Ni et al., 2013*). Grs are also seven-transmembrane receptors (*Figure 9A*) that are distantly related to Ors and

typically mediate gustatory responses (*Clyne et al., 2000*; *Scott et al., 2001*; *Vosshall and Stocker, 2007*).

We started by examining co-receptor expression. In our adult transcriptomes, 68% of neurons highly expressed *Orco* (*Figure 9—figure supplement 1A*). Although highly enriched in Or-positive ORN clusters, we found that *Orco* was expressed in many Ir and Gr clusters (*Figure 9B*; *Figure 9—figure supplement 1B*). With the exception of *Or35a* expressing ORNs, which co-express *Orco* and *Ir76b* (*Couto et al., 2005*; *Fishilevich and Vosshall, 2005*; *Yao et al., 2005*; *Benton et al., 2009*), *Orco* is not thought to be co-expressed with Irs or Grs. Since we observed wide-spread expression of *Orco*, we also evaluated expression of the Ir co-receptors, *Ir8a, Ir25a,* and *Ir76b*. As previously reported (*Silbering et al., 2011*), *Ir8a* expression was restricted to Ir-positive clusters (*Figure 9B*; *Figure 9—figure supplement 1C*). *Ir76b* expression was similarly restricted to Ir-positive clusters, but it displayed low-level expression in some Or-positive clusters (*Figure 9B*; *Figure 9—figure supplement 1E*). However, *Ir25a* was more highly and broadly expressed in some Or clusters in addition to Ir clusters (*Figure 9B*; *Figure 9—figure supplement 1D*). These data indicate that Or and Ir co-receptor transcripts are broadly expressed at the cluster-level.

We next probed if *Orco* and Ir co-receptors were found within the same cells. Based on expression levels (*Figure 9—figure supplement 1G–I*), we considered a co-receptor to be expressed if it met a stringent threshold of $\log_2(CPM+1) \geq 5$. Using this criterion, we found that *Orco* was co-expressed broadly with cells expressing *Ir8a, Ir25a,* and *Ir76b* (*Figure 9C*). In addition to Ir co-receptors, *Orco* was also expressed in a subset of *Gr21a*-positive ORNs (*Figure 9—figure supplement 1F*). The extent of co-expression of these co-receptors may be underestimated due to dropouts or expression below our stringent threshold. Our findings are in line with two recent studies that reveal broad expression and function of Ir and Or co-receptors in *Drosophila* and mosquitos (*Task et al., 2020*; *Younger et al., 2020*).

Finally, we asked if any other sensory receptors are co-expressed with Ors, Grs, or Irs. We queried if the expression of any known Ors/Grs/Irs was correlated with that of the sensory receptors used to annotate adult ORN clusters (*Figure 5F*). Expression of *Gr64a, Gr64f,* and *Gr93a* was correlated with the expression of a subset of Ors (*Figure 9—figure supplement 2A*). Both Gr64a and Gr64f mediate sugar detection (*Dahanukar et al., 2007*; *Jiao et al., 2008*), whereas Gr93a is required for caffeine responses (*Lee et al., 2009*) in gustatory neurons. Although expression of these Grs in the antenna has been described (*Menuz et al., 2014*; *Fujii et al., 2015*), their cell-type specific expression patterns were undefined. We found that *Gr64a* expression was correlated with that of *Or47a* (DM3-projecting ORNs) and *Or9a* (VM3-projecting ORNs), and *Gr93a* expression was most strongly associated with *Or47b* (VA1v-projecting ORNs) expression (*Figure 9—figure supplement 2A–C*). Additionally, *Gr64f* expression was broadly correlated with expression of Ors expressed by basiconic ORNs used to detect food-related odorants, including *Or22a/b* (DM2-projecting ORNs), *Or42b* (DM1-projecting ORNs), *Or59b* (DM4-projecting ORNs), *Or92a* (VA2-projecting ORNs), and *Or9a* (VM3-projecting ORNs) (*Figure 9—figure supplement 2A*). *Gr64f* was also found in a few cells in DL5 and DC2 clusters (*Figure 9D*). The expression of these Grs in Or-positive ORNs suggests potential functions for them in these cells. Together, our transcriptomic data revealed cell-type-specific expression of a number of sensory receptors in adult ORNs.

## Differential expression of genes involved in neuronal communication in distinct adult ORN types

Aside from the receptors and co-receptors, little is known about signaling mechanisms that distinguish distinct adult antennal sensory neuron types. To examine this, we first identified 262 differentially expressed genes between Or- and Ir-positive neurons (*Figure 10—figure supplement 1*). In line with previous reports (*Menuz et al., 2014*; *Mohapatra and Menuz, 2019*), many of the differentially expressed genes included those important for chemosensory behavior, such as sensory receptors, *Sensory neuron membrane protein 1* (*Snmp1*; *Benton et al., 2007*), and *ATPase 8B* (*ATP8B*; *Ha et al., 2014*; *Liu et al., 2014*), as well as TFs and other signaling proteins (*Figure 10—figure supplement 1*). These data indicate that Or- and Ir-positive ORNs may utilize distinct genes for their function.

We next sought to define the molecular and functional characteristics of ORNs at the cell-type-specific level. Specifically, we investigated the expression of genes that regulate neuronal excitability, cell-cell signaling, and synaptic transmission among different ORN types. We identified

differentially expressed genes (adjusted p-value < 0.01 by Mann-Whitney U test) in each of the annotated adult clusters compared to the rest of the clusters and plotted expression of genes that fell into the aforementioned categories.

We found that one neuropeptide previously reported to regulate olfactory sensitivity of DM1 ORNs (*Root et al., 2011*), *short neuropeptide F precursor* (*sNPF*), was differentially expressed in adult ORNs (*Figure 10A*). The fact that *sNPF* was enriched in multiple ORN types suggests that it could have functions beyond DM1 ORNs. While many ion channels/subunits displayed broad expression in ORNs, some exhibited cluster-specific expression (*Figure 10B*). For instance, a cyclic nucleotide-gated ion channel subunit (*Cngl*) was largely expressed by sacculus/arista neurons in addition to DC1 and DM1 ORNs (*Figure 10B*). Furthermore, members of the pickpocket family of sodium channels (*ppk19* and *ppk7*) were expressed in distinct trichoid ORN types, and the voltage-gated calcium channel subunit (*Ca-α1T*) was enriched in V and VA2 projecting ORNs (*Figure 10B*). These data reveal that ORNs express a breadth of ion channels.

Next, we assessed expression of G-protein-coupled receptors (GPCRs) that serve as receptors for neurotransmitters, hormones, and neuropeptides. Many of these receptors had wide-spread expression in multiple neuron types, such as a serotonin receptor (*5-HT2B*), dopamine/ecdysteroid receptor (*DopEcR*), the dopamine receptor (*Dop1R1*), metabotropic GABA receptor (*GABA-B-R1*),

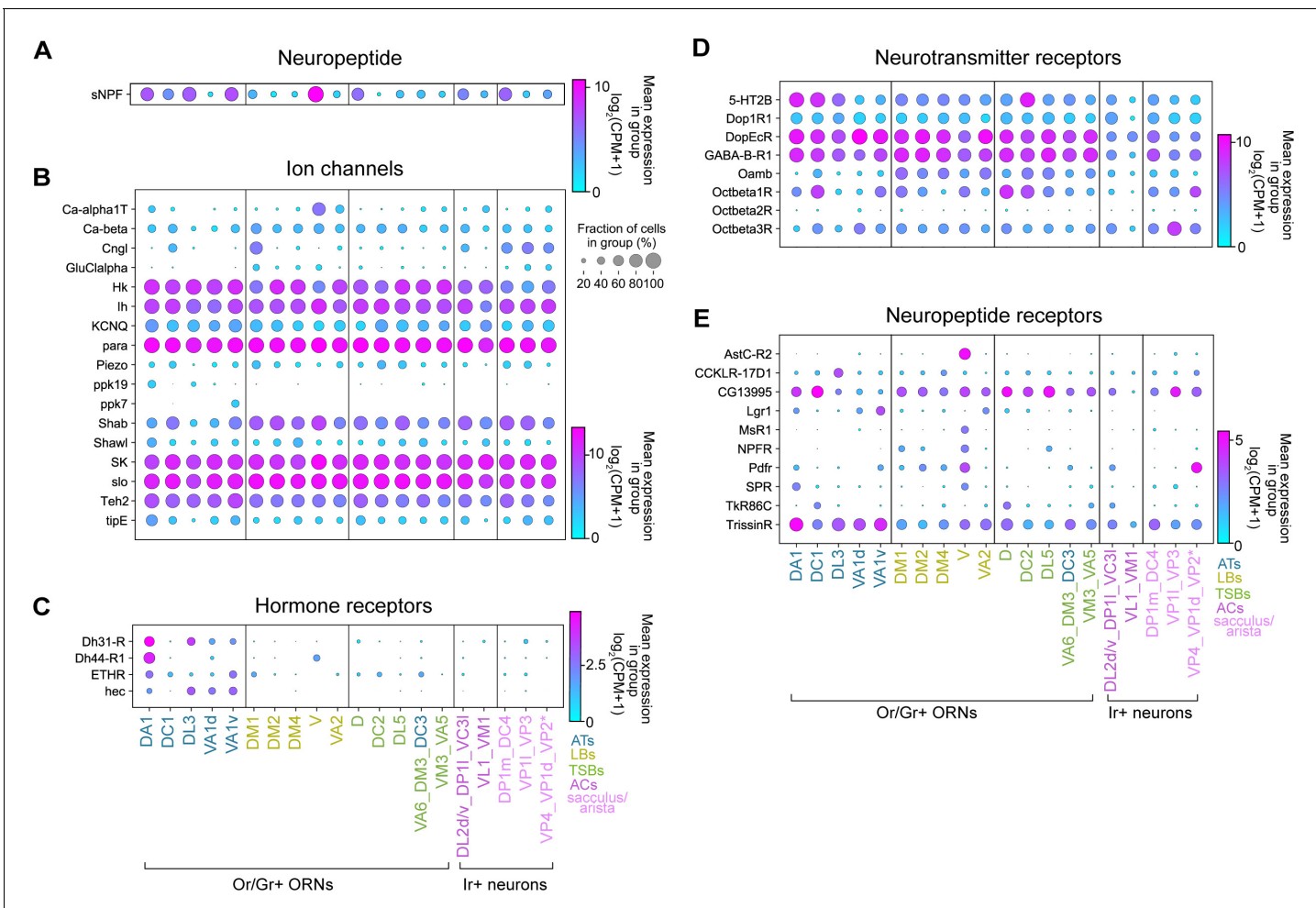

**Figure 10.** Differential expression of genes involved in neuronal communication in distinct ORN types. (A–E) Dot plots depicting differential expression of a neuropeptide (A), ion channels (B), hormone receptors (C), small-molecule neurotransmitter receptors (D), and neuropeptide receptors (E) in annotated adult ORN clusters. Asterisk next to VP2 ORNs indicates that these cells express a Gr and not an Ir.

The online version of this article includes the following figure supplement(s) for figure 10:

**Figure supplement 1.** Differentially expressed genes in Ir- and Or- expressing neurons.

octopamine receptors (*Octβ1R, Octβ3R*), putative neuropeptide receptor (*CG13995*), and neuro-peptide receptor (*TrissinR*) (*Figure 10C–E*). Yet, expression of other GPCRs was restricted to a sub-set of clusters. Hormone receptors were generally expressed at lower levels than other GPCRs but many of these receptors were enriched in ORNs belonging to the trichoid sensillar classes (*Figure 10C*). Conversely, the octopamine neurotransmitter receptor (*Oamb*) was enriched in many basiconic ORN clusters (*Figure 10D*). Similarly, most large basiconic ORNs, as well a few other clus-ters, displayed expression of the *pigment dispersing factor receptor* (*Pdfr*) (*Figure 10E*). Many other neuropeptide receptors were expressed in a few distinct clusters (*Figure 10E*). For example, both the *allatostatin C receptor 2* (*AstC-R2*) and the *myosuppressin1* receptors (*MsR1*) were expressed by V ORNs (*Figure 10E*), suggesting that these ORNs could be modulated by distinct neuropeptides. Taken together, these analyses suggest that discrete ORN types may be modulated by a multitude of signals and exemplify how our adult transcriptomic data can be used as a foundation to investi-gate physiological functions of these sensory neurons.

## Discussion

In this study, we generated high-quality transcriptomes of ~3,100 ORNs from the *Drosophila* third antennal segment at 24 hr APF and in adults. We integrated these data with the transcriptomes of mid-pupal ORNs (*Li et al., 2020*). Our clustering methods separated 24 hr APF and adult transcrip-tomes into 36 and 26 clusters, respectively, and further divided 42 hr APF transcriptomes into 34 clusters. We matched 16 annotated ORN types across all three developmental stages. The fact that we observed fewer clusters than the ~50 glomerular types could be because: (1) not enough cells were profiled for some types; (2) closely related types formed a single cluster (e.g. VM5d and VM5v ORNs at 24 hr APF or VA5 and VM3 ORNs in adults; *Figures 4F*, *5G*); or (3) some Ir-positive neuro-nal types expressed the same receptor and could not be further subdivided transcriptomically (*Figure 5F*; *Figure 5—figure supplement 2M,P*). Our analysis of sensory receptor and marker expression within clusters points to the latter two explanations accounting for the majority of the reduction in the number of clusters compared to the number of neuron types. We were also unable to annotate the antennal thermo- and hygrosensory neurons in our pupal data, possibly because these cells both exist in low numbers in the antenna (*Marin et al., 2020*) and did not express cluster-specific receptors in pupal transcriptomes. Nonetheless, we annotated the majority of clusters at all three stages and found that they represented anatomically and functionally distinct sensory neuron types.

### Single-nucleus RNA-seq can be used to profile cuticle-associated tissues

Single-cell transcriptomics is a powerful tool for discovering novel cell types, revealing type-specific markers, and studying gene expression changes underlying cellular development and maturation (*Li, 2020*). Adult ORNs, like many insect tissues, are tightly associated with the exoskeleton and are difficult to isolate using standard cell dissociation protocols. Nuclei, on the other hand, are much more resistant to mechanical assaults and have been readily profiled using snRNA-seq in multiple systems (*Grindberg et al., 2013*; *Habib et al., 2016*; *Habib et al., 2017*; *Cui et al., 2020*; *Kebschull et al., 2020*; *Lau et al., 2020*). The advantages of snRNA-seq include: (1) tissue can be kept in long-term storage at –80°C prior to processing (*Wu et al., 2019*; *Ding et al., 2020*); (2) snRNAseq reduces cell-type sampling bias (*Denisenko et al., 2020*); and (3) nuclei are more easily captured by microfluidics devices than large and/or fragile cells (*Cui et al., 2020*; *Denisenko et al., 2020*). Underscoring the utility of this protocol, high-quality nuclei can be isolated from frozen tis-sues which allows researchers studying low-abundance cell types to collect and freeze a large amount of tissue over time to ensure that they have an adequate number of nuclei for downstream processing and analyses.

Here, we directly compared snRNA-seq with scRNA-seq of both peripheral and central neurons: 24 hr APF ORNs and adult PNs, respectively. Similar to a previous report in mammalian cortical neu-rons (*Bakken et al., 2018*), we observed a reduction in genes detected in nuclei compared to stage-matched cells. Including intronic reads in our nuclear transcriptomes increased gene detection rates in both neuron types (*Figure 2—figure supplement 1B, C*). Critically, we found that our scRNA-seq and snRNA-seq protocols detected largely overlapping genes (*Figure 2K,L*). Finally, we showed that this method could be used to separate closely-related adult transcriptomes (*Figure 5*). Collectively,

we demonstrated that snRNA-seq is well-suited for profiling previously inaccessible neurons in the adult fly and this protocol can be easily used by other researchers that want to gain transcriptomic access to their tissue of interest (see a detailed protocol in Materials and methods section).

## ORN transcriptomic similarity reflects multiple biological factors

Understanding the molecular underpinnings of neural circuit formation, maintenance, and function requires that we can access the transcriptomes of anatomically and functionally distinct neuron types at developmentally-relevant stages. Here, we created a dataset that will enable these types of analyses by profiling *Drosophila* ORNs at stages corresponding to: (1) initial axon pathfinding (24 hr APF); (2) synaptic partner matching (42 hr APF; *Li et al., 2020*); and (3) sensory function (adult).

Cluster-level comparisons within each newly profiled stage lead to the discovery that type-specific transcriptomes reflect multiple biological factors. During development, similarities in 24 hr APF ORN transcriptomes reflect both their sensillar class and axon trajectories (*Figure 7B*). The abundant differentially expressed transcription factors and cell-surface molecules that we defined in distinct ORN types at both 24 hr and 42 hr APF (*Figure 8*; *Figure 8—figure supplements 1–6*) are excellent candidates to test as regulators of cell-type-specific axon guidance and target selection.

In adults, we annotated olfactory, thermosensory, and hygrosensory neurons originating from all of the sensory structures of the antenna. Similarities in adult transcriptomes not only reflect their sensillar classes, but also sensory stimuli they detect (*Figure 7C*). Our data suggests that neurons that detect similar cues may require analogous genes for their function. Interestingly, some of the food odor-detecting basiconic ORNs expressed the sugar receptor *Gr64f* (*Jiao et al., 2008*), implying that this receptor may have noncanonical functions in these ORN types (*Figure 9D*). Furthermore, the differentially expressed neuronal transmission and signaling genes (*Figure 10*) could provide important insight into the functional properties of distinct ORN types. Together, our developmental and adult analyses underscore that profiling cells at multiple stages can deepen our understanding of their biology.

The main strength of single-cell transcriptomics is the ability to reveal cell-type specific gene expression patterns. By profiling *Drosophila* ORNs at multiple stages, we have generated a rich dataset that can be used as a foundation to reveal cell-type-specific regulators of neuron differentiation, axon targeting, synapse formation, and sensory function. In conclusion, this work, together our accompanying paper profiling PNs at multiple stages (*Xie et al., 2021*), should spur novel and important biological discoveries in the fields of sensory system development and function.

# Materials and methods

**Key resources table**

| Reagent type (species) or resource | Designation | Source or reference | Identifiers | Additional information |
|---|---|---|---|---|
| Genetic reagent (*D. melanogaster*) | nSyb-GAL4 | n/a | RRID:BDSC_51635 | n/a |
| Genetic reagent (*D. melanogaster*) | AM29-GAL4 | *Endo et al., 2007* | n/a | n/a |
| Genetic reagent (*D. melanogaster*) | elav-GAL4 | *Luo et al., 1994* | n/a | n/a |
| Genetic reagent (*D. melanogaster*) | VT033006-GAL4 | *Tirian and Dickson, 2017* | RRID:BDSC_73333 | n/a |
| Genetic reagent (*D. melanogaster*) | UAS-FRTSTOP-FRTmCD8GFP | *Potter et al., 2010* | RRID:BDSC_30125 | n/a |
| Genetic reagent (*D. melanogaster*) | GH146-GAL4 | *Stocker et al., 1997* | RRID:BDSC_30026 | n/a |
| Genetic reagent (*D. melanogaster*) | Act-FRT-STOPFRT-GAL4 | *Pignoni and Zipursky, 1997* | n/a | n/a |
| Genetic reagent (*D. melanogaster*) | UAS-Flp | *Duffy et al., 1998* | n/a | n/a |

*Continued on next page*

*Continued*

| Reagent type (species) or resource | Designation | Source or reference | Identifiers | Additional information |
|---|---|---|---|---|
| Genetic reagent (*D. melanogaster*) | *ey-FLP* | *Chotard et al., 2005* | n/a | n/a |
| Genetic reagent (*D. melanogaster*) | *Or42b-GAL4* | n/a | RRID:BDSC_9972 | n/a |
| Genetic reagent (*D. melanogaster*) | *Or43b-GAL4* | n/a | RRID:BDSC_23895 | n/a |
| Genetic reagent (*D. melanogaster*) | *Or47b-GAL4* | n/a | RRID:BDSC_9984 | n/a |
| Genetic reagent (*D. melanogaster*) | *UAS-lam-GFP* | n/a | BDSC_97378 | n/a |
| Genetic reagent (*D. melanogaster*) | *UAS-unc84-GFP* | *Henry et al., 2012* | n/a | n/a |
| Antibody | Rat anti-Ncad (rat monoclonal) | Developmental Studies Hybridoma Bank | RRID:AB_528121 | (1:40 in 5% normal goat serum) |
| Antibody | Chicken anti-GFP (Chicken polyclonal) | Aves Labs | RRID:AB_10000240 | (1:1000 in 5% normal goat serum) |
| Antibody | Mouse anti-Acj6 (mouse monoclonal) | Developmental Studies Hybridoma Bank | RRID:AB_528067 | (1:40 in 5% normal goat serum) |
| Antibody | Rat anti-Elav (rat monoclonal) | Developmental Studies Hybridoma Bank | RRID:AB_528218 | (1:100 in 5% normal goat serum) |
| Antibody | Mouse anti-Elav (mouse monoclonal) | Developmental Studies Hybridoma Bank | RRID:AB_528217 | (1:100 in 5% normal goat serum) |
| Software, algorithm | ZEN | Carl Zeiss | RRID:SCR_013672 | n/a |
| Software, algorithm | Fiji | National Institutes of Health | RRID:SCR_002285 | n/a |
| Software, algorithm | Illustrator | Adobe | RRID:SCR_010279 | n/a |
| Software, algorithm | STAR 2.5.4 | *Dobin et al., 2013* | RRID:SCR_015899 | https://github.com/alexdobin/STAR |
| Software, algorithm | Htseq 0.11.2 | *Anders et al., 2015* | RRID:SCR_005514 | https://github.com/htseq/htseq |
| Software, algorithm | MARS | *Brbić et al., 2020* | n/a | https://github.com/snap-stanford/mars |
| Software, algorithm | Scanpy | *Wolf et al., 2018* | RRID:SCR_018139 | https://scanpy.readthedocs.io/en/stable |
| Software, algorithm | HDBSCAN | *McInnes et al., 2018* | n/a | https://hdbscan.readthedocs.io/en/latest/how_hdbscan_works.html |
| Software, algorithm | Flymine | n/a | RRID:SCR_002694 | http://www.flymine.org |
| Software, algorithm | Revigo | n/a | RRID:SCR_005825 | http://revigo.irb.hr |
| Software, algorithm | SciPy | n/a | RRID:SCR_008058 | http://www.scipy.org |
| Software, algorithm | NumPy | n/a | RRID:SCR_008633 | http://www.numpy.org |

*Continued on next page*

*Continued*

| Reagent type (species) or resource | Designation | Source or reference | Identifiers | Additional information |
|---|---|---|---|---|
| Software, algorithm | Seaborn | n/a | RRID:SCR_018132 | https://seaborn.pydata.org |
| Software, algorithm | Pandas | n/a | RRID:SCR_018214 | https://pandas.pydata.org |
| Software, algorithm | Scikit learn | n/a | n/a | https://scikit-learn.org/stable |
| Software, algorithm | STRING | n/a | RRID:SCR_005223 | http://string.embl.de |
| Software, algorithm | Cytoscape v3.7.1 | n/a | RRID:SCR_003032 | http://cytoscape.org |

### *Drosophila* stocks and genotypes

Flies were maintained at 25°C on standard cornmeal medium with at 12 hr light-dark cycle. The following fly lines were used in this study: *nSyb-GAL4* (Bloomington *Drosophila* Stock Center, BDSC #51635); *AM29-GAL4* (*Endo et al., 2007*); *elav-GAL4* (*Luo et al., 1994*); *UAS-FRT-STOP-FRT-mCD8-GFP* (*Potter et al., 2010*); *ey-FLP* (*Chotard et al., 2005*); *VT033006-GAL4* (*Tirian and Dickson, 2017*); *Or42b-GAL4* (BDSC #9972); *Or43b-GAL4* (BDSC #23895), *Or47b-GAL4* (BDSC #9984); *GH146-GAL4* (*Stocker et al., 1997*); *UAS-lam-GFP* (BDSC #97378); *UAS-unc84-GFP* (*Henry et al., 2012*); UAS-Flp (*Duffy et al., 1998*); *Act-FRT-STOP-FRT-GAL4* (*Pignoni and Zipursky, 1997*). Note: 48 hr APF and adult cells/nuclei were labeled with *nSyb-GAL4*; however, this driver was not sufficiently expressed at 24 hr APF so cells and nuclei from this earlier stage were labeled using *elav-GAL4*. Also, PN cells were labeled with *GH146-GAL4* and nuclei with *VT03006-GAL4*. For the permanent labeling experiments, *Or-GAL4* lines were crossed with permanent labeling ready flies, *UAS-Flp, Act-FRT-STOP-FRT-GAL4, UAS-CD8-GFP,* and F1 adult brains were dissected and analyzed.

### Immunofluorescence and image acquisition

Fly brains and antennae were dissected and labeled according to previously described methods (*Wu and Luo, 2006*; *Li et al., 2020*). Briefly, brains and antennae were rapidly dissected in PBS and put into ice cold 4% paraformaldehyde in PBS, washed and permeabilized in PBST (1X PBS, 0.3% Triton-X-100), incubated in blocking buffer (5% normal goat serum in PBST), incubated in primary antibody overnight at 4°C, washed in PBST, and incubated in secondary antibody overnight at 4°C. The following primary antibodies were used in this study: rat anti-Ncad (N-Ex #8; at 1:40; Developmental Studies Hybridoma Bank [DSHB]); chicken anti-GFP (1:1000; Aves Labs); mouse anti-Elav (9F8A9; 1:100; DSHB); rat anti-Elav (7E8A10; 1:100; DSHB); mouse anti-Acj6 (1:40; DSHB). Secondary antibodies raised in goat or donkey against mouse, rat, or chicken antisera and conjugated to Alexa Fluor 488/594/647 (Jackson ImmunoResearch) were used at 1:200. Tissue was mounted in SlowFade Gold (Thermo Fisher Scientific). 16-bit images were acquired using a 20x Plan-apochromat (0.8 NA) or 40x Flaur (1.3 NA) objective on a Zeiss LSM 780 (Carl Zeiss) using Zen software (Carl Zeiss) and processed with Fiji (ImageJ; National Institutes of Health).

### Single-cell RNA-seq

Single-cell RNA-seq of ~2,700 *mCD8-GFP* labeled 24 hr APF ORN cells was performed following the protocol we previously described (*Li et al., 2017*). Briefly, *Drosophila* third antennal segments labeled with mCD8-GFP using specific GAL4 drivers were manually dissected. Single-cell suspensions were then prepared. Individually labeled cells were sorted via fluorescence-activated cell sorting (FACS) into single wells of 384-well plates containing lysis buffer using an SH800 instrument (Sony Biotechnology). Full-length poly(A)-tailed RNA was reverse-transcribed and amplified by PCR following the SMART-seq2 protocol. To increase cDNA yield and detection efficiency, we increased the number of PCR cycles to 25. Primer dimer artifacts were reduced by digesting the reverse-transcribed first-strand cDNA using lambda exonuclease (New England Biolabs) (37°C for 30 min) prior to PCR amplification. Sequencing libraries were prepared from amplified cDNA using tagmentation

(Nextera XT). Sequencing was performed using the Novaseq 6000 Sequencing system (Illumina) with 100 paired-end reads and 2 × 12 bp index reads.

## Single-nucleus RNA-seq

Single-nucleus RNA-seq of ~4,200 adult ORN nuclei was performed using the same SMART-seq2 protocol as single-cell RNA-seq with a few key modifications. First, nuclei are labeled with a fluorescent nuclear marker instead of a membrane-bound one. Second, the dissociation and single-nucleus suspension protocol varied greatly from that used for single-cells, detailed below, and was adapted from *Krishnaswami et al., 2016*. Third, cDNA was amplified with 28 PCR cycles (instead of 25) and reads were aligned to exons and introns.

Single-nucleus suspensions were made using the following protocol:

1. Cross desired GAL4 lines with UAS-nuclear-GFP. Do not use *UAS-nlsGFP*, because it does not give a GFP signal after nucleus isolation. We instead use *UAS-unc84-GFP* and *UAS-lam-GFP*.
2. Dissect tissue in cold Schneider's medium, and use P20 pipet (prior to beginning, coat the tip with extraneous tissues) to transfer tissue into 100 µl Schneider's medium in a nuclease-free 1.5 ml tube on ice. Label the tube clearly using permanent marker. **Note**: for tissues that float in the media (e.g. adult antennae), before dissection, prepare three clean dishes: first with 100% ethanol, second and third with Schneider's medium. Rinse the fly in the first dish with 100% ethanol for 5 s, then rinse the fly in the second dish, and dissect in the third dish.
3. After dissection, spin samples down using bench top centrifuge.
4. Fresh: The sample can be processed for nuclei extraction immediately following dissection.
5. Frozen: Alternatively, the sample can be flash-frozen for long-term storage. Seal the 1.5 ml EP tube with parafilm and put into liquid nitrogen for >30 s. Immediately store the sample in –80°C freezer for long-term storage (several months).
6. Prepare fresh homogenization buffer and keep on ice using the using the recipe detailed below.
7. Thaw samples from –80°C on ice if using frozen samples. Centrifuge samples in 100 µl Schneider's medium using bench top centrifuge, discard medium, and add 100 µl homogenization buffer.
8. Optional: if sample pieces are too big, for example, whole body/head, use pellet pestle motor (Kimble 6HAZ6) to grind the sample for 30–60 s on ice.
9. Add 900 µl homogenization buffer, and transfer 1000 µl homogenized sample into the 1 ml dounce (Wheaton #357538). Dounce set should be autoclaved at 200°C > 5 hr.
10. Release nuclei by 20 strokes of loose Dounce pestle and 40 of tight Dounce pestle. Keep on ice. Avoid bubbles.
11. Filter 1000 µl sample through 5 ml cell strainer (35 µm), and then filter sample using 40 µm Flowmi (BelArt #H13680-0040) into 1.5 ml EP tube.
12. Centrifuge for 10 min at 1000 g at 4°C. Discard the supernatant. Do not disturb the pellet.
13. Re-suspend the nuclei using desired amount (we normally use 500–1000 µl) of 1xPBS/0.5% BSA with RNase inhibitor (9.5 ml 1x PBS, 0.5 ml 10% BSA, 50 µl RNasin Plus). Pipet > 20 times to completely re-suspend the nuclei. Filter sample using 40 µm Flowmi into a new 5 ml FACS tube and keep the tube on ice.
14. Nuclei are stickier than whole cells. For users making single-nucleus suspension for the first time, we suggest taking 10 µl of the single-nucleus suspension, stain with Hoechst-33342 (Invitrogen), and check on a cell counter slide to confirm if they are mostly individual nuclei. If nuclei are not sufficiently dissociated, adjust above steps (e.g. increase the number of strokes of the tight pestle when releasing nuclei).
15. Collect nuclei using FACS. Set up the gates using 4 groups of nuclei: WT nuclei, GFP nuclei, WT nuclei + Hoechst stain, GFP nuclei + Hoechst stain. For different fly tissues, there may be more than one band of nuclei with varying Hoechst intensities. They represent different cell types with different nuclei sizes (polyploidy is common for many fly tissues). It is advised to collect different bands of nuclei and check them under microscope. Collect single nuclei into 384-well plates for Smart-seq2, or into a tube for 10X Genomics (not used in this study; details can be sent upon request).

| | Amount | Storage | Item (add in this order) | Final concentration |
|---|---|---|---|---|
| 1 | 10 ml | RT | H$_2$O (nuclease free) | |
| 2 | 0.856 g | RT | Sucrose | 250 mM |
| 3 | 100 µl | 4°C | 1M Tris pH 8.0 | 10 µm |
| 4 | 250 µl | 4°C | 1M KCl | 25 mM |
| 5 | 50 µl | 4°C | 1M MgCl$_2$ | 5 mM |
| 6 | 100 µl | 4°C | 10% Triton-X 100 | 0.1% |
| 7 | 50 µl | –20°C | RNasin Plus (Promega, N2615) | 0.5% |
| 8 | 200 µl | –20°C aliquots | 50x protease inhibitor (Promega, G6521) | 1x |
| 9 | 50 µl | –20°C | 20 mM DTT | 0.1 mM |

## Sequence alignment and preprocessing

Reads were aligned to the *Drosophila melanogaster* genome (r6.10) using STAR (2.5.4) (*Dobin et al., 2013*). Gene counts were produced using HTseq (0.11.2) with default settings except '-m intersection-strict' (*Anders et al., 2015*). Gene counts of scRNA-seq were generated using exonic GTF files. Gene counts of snRNA-seq were generated using both exonic and intronic GTF files, and the two gene count tables were merged for analysis. Low-quality cells/nuclei having fewer than 50,000 uniquely mapped reads were removed. Sequencing depth was normalized across individual cells by re-scaling gene counts to counts per million reads (CPM). All analyses (after *Figure 2*) were performed after converting gene counts to log$_2$ (CPM+1). Non-neuronal cells were filtered out by selecting for the cells with expression of at least two of five neuronal markers (*elav, brp, CadN, nSyb, Syt1*) at log$_2$(CPM+1) ≥ 2. Additionally, we removed neurons that expressed auditory receptor neuron-specific genes (*iav, nompC, CG9492, CG10186, fd3F*) in 24 hr APF and adult transcriptomes (*Figure 4—figure supplement 1*; *Figure 5—figure supplement 1C*), as well as, *acj6* negative neurons from analyses in *Figure 4* onward.

## Dimensionality reduction and HDBSCAN clustering

Genes for dimensionality reduction were selected using previously described over-dispersion methods (*Li et al., 2020*) or by calculating differentially expressed genes between all of the time points (*Figure 1D*), or within a single time point (*Figures 4G*, *5F*, *6A, B*). Note the differentially expressed genes that were used for dimensionality reduction of each individual time point (*Figures 4*, *5*) and all time points in *Figure 6* were identified at 42 hr APF and combined with known antennal olfactory receptors from *Grabe et al., 2016* for a total of 335 genes. Principal component analysis (PCA) followed by t-distributed Stochastic Neighbor Embedding (t-SNE; *van der Maaten and Hinton, 2008*) was used to jointly visualize cells/nuclei from all stages. Briefly, we obtained two-dimensional projections of cell population by first reducing the dimensionality of the gene expression matrix and then projecting these genes in to the t-SNE space. We used Uniform Manifold Approximation and Projection (UMAP; *McInnes et al., 2018*) for visualization of cells/nuclei at their individual stages. UMAP was used for stage-specific embedding because it better preserved the global structure of the data and resulted in clearer depictions of the clusters at each time point. The fewer than 100 cells per time point that did not associate with a cluster (from MARS analysis) were not depicted on the tSNE or UMAP plots to make these plots easier to decipher. The following HDBSCAN settings were used to classify cells into clusters in an unbiased manner min_cluster_size = 6 and min_samples = 4.

## MARS clustering

We calculated differentially expressed genes among 42 hr APF ORN types that we identified in *Li et al., 2020*. We combined these genes with known antennal sensory receptors (*Grabe et al., 2016*), resulting in a total of 335 genes. We used this set of genes to confirm ORN types at 42 hr APF, and to identify ORN types at 24 hr APF and adult. To cluster each dataset, we applied MARS (*Brbić et al., 2020*)—a meta-learning approach for cell type discovery. MARS leverages annotations of previously annotated datasets to better separate cell types in an unannotated dataset. We first applied MARS to re-annotate 42 hr APF ORN types. The annotations agreed well with our previously

identified types (*Li et al., 2020*), and we used them to annotate 24 hr APF. We ran MARS seven times with different random initializations (compared to nine times on 42 hr APF transcriptomes) and neural network architectures to increase our confidence in the discovered clusters, and combined annotations from the different runs. A cluster was approved by calculating differentially expressed genes in that cluster and checking whether genes agree with known markers and/or are expressed uniquely in that cluster compared to other cells not assigned to the cluster. After annotating the 24 hr APF data, we used these annotations in combination with 42 APF annotations to guide the clustering of adult neurons. For adult annotations, we proceeded in the same way as when annotating 24 hr APF and ran MARS ten times followed by validating each cluster using marker genes.

## Transcriptome similarity analysis

To compare the transcriptome similarity between 24 hr APF, 42 hr APF, and adult clusters, we computed a set of unbiased 416 differentially expressed genes across all three developmental stages. For each cluster, we calculated expression profile of differentially expressed genes by taking the average across all cells within that cluster. We then calculated Pearson correlation coefficient of selected genes between all pairs of MARS clusters at each stage. For the transcriptome similarity analysis in *Figure 4B*, we used all clusters discovered by MARS.

## Gene ontology and STRING analysis

Gene Ontology (GO) analysis was used to characterize transcriptome changes in multiple cases. We generated gene lists for GO analysis by comparing the two groups using differential expression analysis (Mann-Whitney U test). We identified roughly equal numbers of differentially expressed genes that reached a significance level of p-value $\leq 10^{-5}$ (after Bonferroni adjustment for multiple testing) for each comparison group and uploaded these gene lists into Flymine for GO analysis (*Lyne et al., 2007*). Redundant GO terms were removed using REVIGO (*Supek et al., 2011*). We report the p-value of enrichment for each term. For adult GO analysis in *Figure 2* we removed mitochondrial gene and odorant binding protein contamination prior to performing the analysis.

The STRING network was made using the top 100 differentially expressed genes (by fold change) from 24 hr APF ORN nuclei and adult nuclei into the STRING database search portal. Nodes that did not connect to the network were not displayed, and edges (links between genes) were generated by their reported interactions and corresponding confidence scores and plotted in Cytoscape (v3.7.1). Clusters were created and labeled using the GO terms that they were associated with and colored based on their on their fold change which was calculated by $\log_2[(\text{mean}_{adult} +0.1) / (\text{mean}_{24h\ APF} + 0.1)]$. Only clusters whose genes were found in the REVIGO reduced GO terms were displayed.

## Generation of dendrograms for 24 hr APF and adult transcriptomes

Dendrograms for 24 hr APF and adult data were derived from hierarchical clustering of annotated clusters at each stage. Pair-wise comparisons between all annotated clusters at each stage were performed to generate a list of differentially expressed genes. This resulted in 437 differentially expressed genes at 24 hr APF and 936 differentially expressed genes in adult clusters. Average gene expression was then computed for each cluster and differentially expressed genes were used for dimensionality reduction at their respective stages, followed by hierarchical clustering using the cluster map function of seaborn based on the Euclidean distance between each cluster.

## Manual (marker-based) matching and annotating of ORN types

ORN clusters were manually matched using a combination of differentially expressed marker genes and sensory receptor (Ir, Gr, Or) expression. Marker genes were identified using our previously published cluster-specific genes at 42 hr APF (*Li et al., 2020*) or via a Mann-Whitney U test to find genes that are highly expressed in one (or a few) clusters compared to the rest. We searched for either a single gene or a set of genes whose expression was unique to a single cluster at both 24 hr APF and 42 hr APF. If these marker genes were expressed in one cluster at both time points, we considered these clusters to be the same ORN type. We used olfactory receptor expression to match some clusters from 24 to 42 hr APF. Sensory receptors were exclusively used to annotate adult clusters and to match them to their corresponding 42 hr cluster. Genes used to annotate, and match clusters are summarized in dot plots in *Figures 4*, *5*.

## Automatic matching of ORN types

To automatically identify the same ORN types across different developmental stages, we first calculated differentially expressed genes in MARS clusters at each stage. We started by matching clusters between 24 and 42 hr APF, then we matched clusters between 42 hr and adult APF, and finally used matching between adult and 24 hr APF to confirm matches. To match clusters between two stages, we computed the similarity score of differentially expressed genes between all pairs of clusters using Jaccard similarity index, defined as the ratio of the number of elements in the intersection of two gene sets and the number of elements in the union of the gene sets. Two-way matches occurred when cluster X in one stage had the highest similarity to cluster Y in another stage, and the opposite also held true where cluster Y had the highest similarity to cluster X. One-way matches occurred when cluster X at one stage has the highest similarity to cluster Y at another stage, but cluster Y is most similar to another cluster. Since each cluster has a one-way match by design of the approach, we relied on one-way matching exclusively for those clusters whose transcriptomes are intermingled (inseparable) between 24 and 42 hr APF (*Figure 7—figure supplement 1A*).

## Differential gene expression analysis

We identified differentially expressed genes in developing (*Figure 8*) and adult clusters (*Figure 10*) by performing a Mann-Whitney U test on each cluster (at each stage) compared to the rest of the clusters within that stage. p-Values were adjusted using via Benjamini-Hochberg procedure. Genes with an adjusted p-value<0.01 were used in follow-up analyses. We used curated lists to identify specific gene categories within differentially expressed gene lists. The TF list originated from the FlyTF database (*Pfreundt et al., 2010*) and the CSM list was from *Kurusu et al., 2008*. These lists were manually curated to remove redundancies and spurious annotations according to FlyBase annotations. All other gene categories (found in *Figure 10*) were based on gene groups obtained from FlyBase.

## Quantification and statistical analysis

All RNA-seq data analysis was performed in Python using NumPy, SciPy, seaborn, pandas, scikit-learn, Scanpy (*Wolf et al., 2018*) and custom scRNA-seq analysis modules (*Brbić et al., 2020*; *Li et al., 2017*). Sequencing reads and pre-processed sequencing data are available in the NCBI Gene Expression Omnibus (GSE162121). Custom analysis code is available at: https://github.com/colleenmclaughlin/ORN_seq/ (*McLaughlin, 2020*; copy archived at swh:1:rev:9e336a9ac1c7af0344d8ebac90c1b631688ed4eb).

## Acknowledgements

We are grateful to Daniel Pederick, Jan Lui, Chuanyun Xu, David Luginbuhl, and Zhuoran Li for helpful discussions about this work and comments on the manuscript. We thank all members of the Luo lab for support and advice. We thank Dr. Gilbert L Henry for sharing UAS-unc84-GFP flies. We thank the Bloomington *Drosophila* Stock Center, the Vienna *Drosophila* Resource center, and the Developmental Studies Hybridoma Bank for fly lines and antibodies. We are grateful to Mary Molacavage for administrative support. Finally, we thank James Ferguson for help making the diagram in *Figure 1A*.

## Additional information

### Funding

| Funder | Grant reference number | Author |
| --- | --- | --- |
| National Institutes of Health | R01 DC005982 | Liqun Luo |
| National Institutes of Health | K99 AG062746 | Hongjie Li |
| Damon Runyon Cancer Research Foundation | Howard Hughes Medical Institute Fellow_DRG 2390-20 | Colleen N McLaughlin |
| Howard Hughes Medical Institute | Investigator | Liqun Luo |

| | | |
|---|---|---|
| Jane Coffin Childs Memorial Fund for Medical Research | Fellow | Justus M Kebschull |
| National Science Foundation | OAC-1835598 (CINES) | Jure Leskovec |
| National Science Foundation | OAC-1934578 (HDR) | Jure Leskovec |
| National Science Foundation | CCF-1918940 (Expeditions) | Jure Leskovec |
| National Science Foundation | IIS-2030477 (RAPID) | Jure Leskovec |
| Stanford Data Science Initiative | | Jure Leskovec |
| Chan Zuckerberg Biohub | Investigator | Jure Leskovec Stephen R Quake |
| Stanford Wu Tsai Neuroscience Institute | Fellow | Hongjie Li |
| Wu Tsai Neurosciences Institute | Neuro-omics program | Liqun Luo Stephen R Quake Jure Leskovec |

The funders had no role in study design, data collection and interpretation, or the decision to submit the work for publication.

## Author contributions

Colleen N McLaughlin, Conceptualization, Resources, Data curation, Software, Formal analysis, Validation, Investigation, Visualization, Methodology, Writing - original draft, Writing - review and editing; Maria Brbić, Data curation, Software, Formal analysis, Visualization, Methodology, Writing - original draft, Writing - review and editing; Qijing Xie, Investigation, Writing - review and editing; Tongchao Li, David Vacek, Anthony Xie, Jiefu Li, Investigation; Felix Horns, Resources, Investigation; Sai Saroja Kolluru, Justus M Kebschull, Robert C Jones, Jure Leskovec, Resources; Stephen R Quake, Funding acquisition; Liqun Luo, Conceptualization, Resources, Supervision, Funding acquisition, Writing - original draft, Writing - review and editing; Hongjie Li, Conceptualization, Resources, Data curation, Formal analysis, Supervision, Investigation, Methodology, Writing - original draft

## Author ORCIDs

Colleen N McLaughlin https://orcid.org/0000-0002-7518-8920
Qijing Xie http://orcid.org/0000-0002-0997-8326
Felix Horns http://orcid.org/0000-0001-5872-5061
Jiefu Li http://orcid.org/0000-0002-0062-4652
Robert C Jones http://orcid.org/0000-0001-7235-9854
Stephen R Quake http://orcid.org/0000-0002-1613-0809
Liqun Luo https://orcid.org/0000-0001-5467-9264

## Decision letter and Author response

Decision letter https://doi.org/10.7554/eLife.63856.sa1
Author response https://doi.org/10.7554/eLife.63856.sa2

# Additional files

## Supplementary files

- Transparent reporting form

## Data availability

All data generated or analyses used in this study are deposited in the requisite repositories. Sequencing data are uploaded to Gene Expression Omnibus (GSE162121). Python code for performing analyses and generating figures has been uploaded to GitHub (https://github.com/colleen-mclaughlin/ORN_seq/; copy archived at https://archive.softwareheritage.org/swh:1:rev:9e336a9ac1c7af0344d8ebac90c1b631688ed4eb/).

The following dataset was generated:

| Author(s) | Year | Dataset title | Dataset URL | Database and Identifier |
|---|---|---|---|---|
| McLaughlin CN, Brbić M, Xie Q, Li T, Horns F, Kolluru SS, Kebschull JM, Vacek D, Xie A, Li J, Jones RC, Leskovec J, Quake SR, Luo L, Li H | 2021 | Single-cell transcriptomes of developing and adult olfactory receptor neurons in Drosophila | https://www.ncbi.nlm.nih.gov/geo/query/acc.cgi?acc=GSE162121 | NCBI Gene Expression Omnibus, GSE162121 |

The following previously published dataset was used:

| Author(s) | Year | Dataset title | Dataset URL | Database and Identifier |
|---|---|---|---|---|
| Li H, Li T, Horns F, Xie Q, Li J, Xu C, Wu B, Kebschull JM, McLaughlin CN, Kolluru SS, Jones RC, Vacek D, Xie A, Luginbuhl D, Quake SR, Luo L | 2020 | Single-cell Transcriptomes Reveal Diverse Regulatory Strategies for Olfactory Receptor Expression and Axon Targeting | https://www.ncbi.nlm.nih.gov/geo/query/acc.cgi?acc=GSE143038 | NCBI Gene Expression Omnibus, GSE143038 |

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
