## [Decision Letter]

**Acceptance summary:**

Your paper provides a comprehensive transcriptional dataset from single OSNs while they develop and defines many of the genes that contribute to sensory, axon projection and connectivity properties. We believe that this will be a great resource for the community.

**Decision letter after peer review:**

Thank you for submitting your article "Single-cell transcriptomes of developing and adult olfactory receptor neurons in *Drosophila*" for consideration by *eLife*. Your article has been reviewed by three peer reviewers, and the evaluation has been overseen by a Hugo Bellen as the Reviewing Editor and K VijayRaghavan as the Senior Editor. The following individual involved in review of your submission has agreed to reveal their identity: Scott Barish (Reviewer #2).

The reviewers have discussed the reviews with one another and the Reviewing Editor has drafted this decision to help you prepare a revised submission.

The three reviewers feel that novelty is lacking but they all three are in agreement that this will be a useful resource. If you can address most of the issues raised, we will send the paper back to the reviewers for a new assessment. We suggest that you resubmit this as a resource paper unless you have good reasons to object.

(Also: It is important to cite and compare results to whole antennae transcriptomes published by others (Mohapatra and Menuz, 2019; Li et al., 2016; Barish et al., 2017; Menuz et al., 2014). Please include these)

Reviewer #1:

This paper from McLaaughlin et al., builds upon the previous paper from the Luo lab in Current Biology (2020) on single cell RNA profiling from developing Olfactory receptor neuron (ORN) populations in the *Drosophila* olfactory system. In this paper, the authors have expanded on the previous paper's findings, by providing transcriptional profiles from single ORNs from mid-pupal and adult stages as additional developmental time points. They were able to identify majority of the ORN classes based on gene clusters. In addition, building on their previous data on the ORN clusters from pupal stages, they were able to provide developmental trajectories for majority of the ORN classes based on transcriptional profiles and how they change over developmental time. The study identified that each ORN only expresses a single OR during its development. This is in contrast to mammalian ORNs, which initially express multiple OR genes which later consolidate into a single OR genes per ORN class. They also were able to identify class-specific transcriptional signatures for adult ORN classes as well as developmental trajectories of transcriptional profiles within ORN lineages. In general, this paper, together with the group's previous papers on scRNA profiling from developing ORN and PNs as well as adult PNs, provides a comprehensive transcriptional dataset from developing ORNs that highlight the genes that contribute to sensory, functional and morphological properties of different ORN classes. I think the paper is suitable for possible publication in e*Life* if the points below are addressed.

1) There is a preprint from the Benton lab, which is currently under review, on transcriptional profiling of ORN classes using the DamID-RNA polymerase II occupancy in 7 ORN classes. Did the authors look at this? There seems to be some inconsistencies regarding single OR gene expression, source of odorant binding protein expression (support cell versus ORNs)

2) The authors were able to identify ORN class specific expression from approximately 80% of the 50 ORN classes. They reason that this might be due to either not enough cells sampled from the remaining ORN classes, co-expression of IR genes in coeloconic ORNs that might interfere with class-specific clustering of these ORNs, or inability to segregate clusters for closely related ORNs. Since this is a follow up study that mainly incorporates adult ORN data, it would be better if it was providing a complete transcriptional profile of all adult ORNs. I am wondering if increasing the total number of cells sequenced would help resolve the problem by providing statistical power. I am not sure how long it would take to do this though.

3) Since the same group also have access to data from PN populations, are there any genes (cell surface receptors, transcription factors, neurotransmitter versus neurotransmitter receptors) expressed in ORN-PN pairs targeting the same glomerulus? For example, are complementary heterophilic or homophilic cell surface receptors are expressed in the ORN-PN pairs, or neurotransmitters expressed in ORN class and the complementary expression of its receptor in PNs? I think such a cross comparison can yield many important insights into the circuit assembly programs.

4) A small number of OR genes shows co-expression with other OR genes are expressed in more than one class of ORNs. Is there anything in common among the OR genes that are co-expressed. It is clear that some are recent duplication events where the genes are repeated in a single genomic locus (Or65a/b/c or Or22a/b).But others are not so close on the genome. Is there anything in common among these ORs? For example, is there any amino acid sequence similarity suggestive of more distant duplication and translocation of the gene to another region in the genome that retained some of the same regulatory elements? Or co-expression seen in ORNs found in the same sensilla type, which might indicate regulatory similarities?

5) The authors make the conclusion that a given OR is expressed in the same ORN from birth until adulthood, in contrast to mammalian ORNs which initially express multiple OR genes but then consolidate to express a single one. They use both their RNAseq data and a permanent genetic labeling to show this. In the text they mention that generally most of the ORN that express a single OR gene innervate a single glomerulus even when they perform the permanent genetic labeling. However, in Figures 3E and the supplement, they show images where permanent labeling leads to innervation of some weak secondary glomeruli by a single OR-GAL4. I could not find an explanation for this in the main text. Can the authors elaborate on these? Is this a GAL4 artifact? Or a real indication that these OR genes initially are also expressed in other ORNs but are later consolidated to a single ORN class?

6) It would be good to get an expanded clustered heat maps and lists for all ORN classes and the expression of specific gene groups, like cell surface receptors, transcription factors, etc.

7) Some of the wording on the figures, especially in heatmaps etc. are extremely small. Please make them bigger in addition to providing a separate table with expression levels.

8) For Figure 1B, can you provide the names of the ORs co-expressed that are shown on the heatmap?

Reviewer #2:

In this work, the authors provide a thorough analysis of single cell sequencing from developing *Drosophila* ORN tissue. This is the first comprehensive single cell RNA seq analysis from multiple developmental stages of ORN development and undoubtedly a valuable resource for the field. Overall the paper is well written and the experiments are rigorous. However, I have some significant concerns.

1) Novelty: While this is the first single-cell RNA seq paper, it is not the first to conduct RNA-seq on ORN tissue across development. Notably, these studies are not cited and the results are therefore not compared.

2) There are very few new conclusions to be drawn from this paper that haven't been identified in other studies. It was already well known that ORNs only express one receptor in their lifetime unlike mice. The few genes that are novel to ORN development are not discussed in any detail. Only forkhead gets a passing mention in the text

3) Even though few novel genes are mentioned, there is also almost no discussion of known genes either. Do the authors see expression of wiring genes like robo, dscam, and DIP/Dprs? What about TFs known to control ORN development? Atonal, rotund, amos, ect.? There are several TFs known to maintain OR expression in the adult antenna. Do the authors see expression that matches these results? Again, several studies are not cited here.

Reviewer #3:

This study generated single-cell transcriptomes from neurons of the *Drosophila* antenna at 24h APF and adult antennae. It uses computational techniques to match most (60% and 80%, respectively) transcriptional clusters to defined neuronal types. The authors have earlier published (Li et al., 2020) single-cell transcriptomes at 42h APF, and those data are compared to the new data in the current study. It's not clear that the present study produces a major conceptual advance, but it describes the application of single nucleus sequencing to *Drosophila* and provides a resource that should be very useful to many in the field.

1) Do ORNs in the same sensillum (for example ab2A and ab2B) share more similar transcriptomes?

2) It would be helpful to compare the results of snRNAseq with results from previous RNAseq of the entire antenna. This comparison might reveal if there is any bias imposed by the snRNAseq methods used here. Are fewer Ors identified in the present study? If so, are there any commonalities among those that are missing from the snRNA seq? Are there striking differences in the relative abundances between the two methods? Was Or85c observed in whole antennal RNAseq?

3) Figure 1M: Why does the gene ontology analysis identify "chemotaxis" as a top term at 24h but not at the adult stage?

4) The authors should clarify here the limitations of the analysis. "Cluster-specific Or and Gr expression allowed us to match 13 transcriptomic clusters with their glomerular types." There should be a clearer explanation here for why the number is not higher. Likewise, the authors should explain more explicitly why transcriptomes of some ORNs are indistinguishable at the adult stage – they should express different receptors. It would be helpful to provide a list of clusters containing multiple neurons, indicating for each cluster which neurons map to it.

5) The studies of the requirements for G proteins (for example Kain et al.) need to be stated more accurately.

6) Figure 3C. The chord plot shows co-expression of Or85c and Or85b, which are located near each other in the genome. Is there a tendency for other genes that are near each other to be co-expressed?

7) The detection of mitochondrial transcripts in ORN nuclei is said to result from dissociation artifacts. Obps are found here to be broadly detected in ORN nuclei. Obps are expressed at extremely high levels in accessory cells. Could their detection in ORN nuclei be artifactual as well?

8) Please state more explicitly how many single nuclei, and how many single cells, are discarded from the analysis, and the reasons for discarding them.

9) There is some concern about the use of intronic reads. In Figure 5C, for example, Ir41a is reported to be expressed in many neurons. Ir41a is a very long gene. Could its apparent expression represent noise resulting from using intronic reads?

---

## [Author Response]

We thank the editors and reviewers for their appreciation of our study and for their candid critiques. In addressing their feedback, we have substantially improved the quality, utility, and scholarship of our manuscript. The number of total figures and supplements has expanded from 20 to 30. We have also substantially streamlined our manuscript to improve the clarity and accessibility to a general audience. We will also take the editors’ and reviewers’ suggestion to resubmit this manuscript as a resource. Below, we first summarize major changes to the manuscript, and then provide a point-by-point response to specific comments from the editors and reviewers.

Major Additions:

1) We have added a large amount of new analyses on differential expression of genes in different neuron types across development, with an emphasis on transcription factors and cell-surface molecules during development. These analyses are presented in the new Figure 8 and Figure 8—figure supplements 1–6. We have also expanded on analyses of differentially expressed genes involved in neuronal communication and signaling in adult neurons, summarized in an expanded Figure 10 and Figure 10—figure supplement 1. We described many differentially expressed transcription factors and cell surface proteins in our developing ORNs. These additions have substantially improved the new biological insights that can be obtained from our dataset. For instance, one can now easily identify transcription factors and cell surface molecules that display cell-type specific expression, as well as expression within a specific sensillar class during development. These genes are not only excellent candidate regulators of ORN development and wiring but can also be used to generate driver lines for genetic access to individual ORN types. Our analyses of adult transcriptomes have also revealed type specific expression of neuronal signaling genes and neuromodulators, which could also serve as good candidates for functional investigations.

2) We have substantially expanded our analysis of co-expression of olfactory coreceptors in Figure 9 and Figure 9—figure supplements 1–2. We quantified the extent of such co-expression from our unique, adult single-cell transcriptome data. Together with two new studies deposited in bioRxiv during the initial review of this manuscript (Task et al., *bioRxiv* https://doi.org/10.1101/2020.11.07.355651; Younger et al., *bioRxiv* https://doi.org/10.1101/2020.11.07.368720), these studies reveal that olfactory receptor co-expression in insects (including flies and mosquitos) is much more substantial than previously thought. We have also systematically examined co-expression of specific olfactory receptors, validating previously documented co-expression of receptors (originating closely linked genomic loci). Additionally, we documented cell-type-specific co-expression of three gustatory receptors in specific ORN types, including two sugar receptors Gr64a and Gr64f. These analyses are summarized in the new Figure 9—figure supplement 2.

3) We have expanded our citations (our reference list expanded from 113 to 126). In particular, we have extensively cited previous bulk RNA-seq studies in multiple relevant places throughout the paper (including lines: 64; 65; 204; 335; 339; 340; 488; 498; 595). We also compared our single-cell transcriptome data with bulk RNA-seq in a number of figures (Figure 2—figure supplement 1O; Figure 5C; Figure 5—figure supplement 1 A; Figure 8; Figure 9; Figure 10).

4) To make room for addition of new data analyses without excessively lengthening the manuscript, we have removed less essential figure panels and associated text from the original submission. Specifically, previous Figure 1D and E have been moved to supplement; images of antennal lobes with additional permanent labeling have also been moved to supplement and streamlined for clarity; Figure 4B has been moved to supplement; Figure 5D has been removed; Figure 7C–G (and Figure 7—supplement 1A) have been removed; most of the panels in the former Figure 8 (now Figure 9) have been re-worked; and all of the panels in the former Figure 9 (now Figure 10) were replaced.

Reviewer #1:This paper from McLaaughlin et al., builds upon the previous paper from the Luo lab in Current Biology (2020) on single cell RNA profiling from developing Olfactory receptor neuron (ORN) populations in the *Drosophila* olfactory system. In this paper, the authors have expanded on the previous paper's findings, by providing transcriptional profiles from single ORNs from mid-pupal and adult stages as additional developmental time points. They were able to identify majority of the ORN classes based on gene clusters. In addition, building on their previous data on the ORN clusters from pupal stages, they were able to provide developmental trajectories for majority of the ORN classes based on transcriptional profiles and how they change over developmental time. The study identified that each ORN only expresses a single OR during its development. This is in contrast to mammalian ORNs, which initially express multiple OR genes which later consolidate into a single OR genes per ORN class. They also were able to identify class-specific transcriptional signatures for adult ORN classes as well as developmental trajectories of transcriptional profiles within ORN lineages. In general, this paper, together with the group's previous papers on scRNA profiling from developing ORN and PNs as well as adult PNs, provides a comprehensive transcriptional dataset from developing ORNs that highlight the genes that contribute to sensory, functional and morphological properties of different ORN classes. I think the paper is suitable for possible publication in eLife if the points below are addressed.1) There is a preprint from the Benton lab, which is currently under review, on transcriptional profiling of ORN classes using the DamID-RNA polymerase II occupancy in 7 ORN classes. Did the authors look at this? There seems to be some inconsistencies regarding single OR gene expression, source of odorant binding protein expression (support cell versus ORNs)

The Benton lab applied the Targeted DamID (TaDa) method to Ir-expressing ORNs to study molecular mechanisms that control ORN development. The authors showed the feasibility of this TaDa profiling method for rare Ir-expressing ORNs, and identified one transcription factor, Pdm3 (previously was shown to control ORN axon guidance and Or expression by the Carlson’s group, Tichy et al., J Neurosci 28:7121, 2008) and one cell surface protein, fmi, in controlling the ORN cell fate and wiring. Since they are taking a bulk approach, their strength is to detect low-expressed Ir genes or very rare cells. They profiled 7 Ir-expressing ORN types (Ir64a, Ir40a, Ir75a, Ir75b, Ir75c, Ir84a, and Ir31a). Our single-cell approach can profile many more ORN types at one time but cannot recover some rare populations, e.g. Ir84a and Ir31a. We did cover other 5 other Ir-expressing ORNs and did see consistent results. For example, they used Ir64a-GAL4 to label ORNs targeting to DC4 and DP1m glomeruli, and we have mapped Ir64a-positive cells to these two ORN types.

The DamID paper only focused on adult Ir-expressing neurons, and we have profiled both development and adult ORNs. These two studies will complement each other to provide useful resource for studying both ORN development and physiology. Since these two methods are very different and have focused on largely non-overlapping cells, it is currently difficult to perform detailed comparison of these two datasets. See reviewer 3 point 7 below for odorant binding protein expression.

2) The authors were able to identify ORN class specific expression from approximately 80% of the 50 ORN classes. They reason that this might be due to either not enough cells sampled from the remaining ORN classes, co-expression of IR genes in coeloconic ORNs that might interfere with class-specific clustering of these ORNs, or inability to segregate clusters for closely related ORNs. Since this is a follow up study that mainly incorporates adult ORN data, it would be better if it was providing a complete transcriptional profile of all adult ORNs. I am wondering if increasing the total number of cells sequenced would help resolve the problem by providing statistical power. I am not sure how long it would take to do this though.

We would first like to clarify that we were able to annotate ~80% of the adult transcriptomic clusters as belonging to 31 receptor neuron types. The reviewer correctly listed several contributing reasons why we were not able to annotate transcriptomes for all ORN types. Other reasons include the large dynamic range of expression levels for type-specific genes, such as *Ors* and *Irs*. For example, Menuz et al. (PLoS Genetics, 10:e1004810, 2014) reported that the levels of canonical *Ors* varied over a 170-fold range from their bulk RNAseq data. There is also a large variation in the number of receptor neurons belonging to each distinct neuron type. For instance, the number of ORNs innervating each glomerulus can vary over a 6-fold range (Grabe et al., Cell Reports 16:3401-3413, 2016). So, neurons that exist in low numbers and have relatively low receptor expression, such as DA3 ORNs (Grabe et al., 2016; Menuz et al., 2014), are challenging to annotate. We expect that our 1,891 single-cell transcriptomes in the adult stage contains the majority of ORN types, but because of the above reasons, we could not detect type-specific markers, or have insufficient number of cells for some of the rare types, to decode more transcriptomes.

While further increasing the sample size will surely generate more data, there is a diminished return in decoding new types because we will preferentially get more abundant cell types and we will still have difficulty in detecting type-specific genes that have low expression levels. The better approach would be to use GAL4 lines that are specific to rare populations for FAC sorting, so that sequencing can be focused on those rare types. Indeed, we are aware of groups who are doing just that.

We note that 1,891 single-cell transcriptomes from FACS-SMARTseq2-based approach (sequenced to a depth of 1 million reads per cell) is considered fairly large scale. (For example, the SMARTseq2-based sequencing we published on mouse serotonin neurons in 2019 was based on 999 cells; see Ren et al., *eLife* 8:e49424; another recent study was based on 1,211 cells sequenced by Smart-seq2 for classifying spiny projection neurons in the mouse striatum: Stanley et al., Neuron 105:688, 2000.) We probably need to sequence several fold more cells in order to decode a substantial number of additional ORN types. We also note that while it is true that this is a “follow up” study for ORN singlecell RNAseq to our 42h dataset (by including both 24h and adult), this is the first time type-specific adult sensory neuron transcriptomes has been reported. The adult transcriptomes are much harder to obtain and we had to develop new (single-nucleus RNA-seq) methods to obtain high-quality transcriptomes. (We have sent protocols described here to many laboratories, including those who work on other insects such as mosquitos and have included a detailed protocol for dissociating nuclei in the Materials and methods section of this manuscript.) And decoding single-cell transcriptomes from 80% of adult clusters should be a substantial advance. Beyond ORNs, we were also able to obtain transcriptomes of thermo- and hygro-receptor neurons. Finally, we would like mention that the preprint of this manuscript in *bioRxiv* generated considerable excitement in researchers studying adult sensory neurons, and we have already shared our adult dataset with some of these labs.

We hope that sequencing more cells will not be a requirement for this paper to be published in *eLife*. Rather, we have focused our revision on analyzing the existing data and comparison with previous literature, following many useful suggestions of the reviewers. Thank you for your understanding.

3) Since the same group also have access to data from PN populations, are there any genes (cell surface receptors, transcription factors, neurotransmitter versus neurotransmitter receptors) expressed in ORN-PN pairs targeting the same glomerulus? For example, are complementary heterophilic or homophilic cell surface receptors are expressed in the ORN-PN pairs, or neurotransmitters expressed in ORN class and the complementary expression of its receptor in PNs? I think such a cross comparison can yield many important insights into the circuit assembly programs.

Thank you for this suggestion. Indeed, comparing ORN and PN transcriptomes during development to identify wiring specificity genes is a major incentive for us to carry out these studies. In our revised manuscript, we have listed differentially expressed transcription factors and cell-surface molecules in ORNs, as summarized in Major Addition 1 above. As requested, in the new Figure 8—figure supplement 5 and 6, we listed cell-surface molecules that are differentially expressed by 6 matching pairs of ORNs and PNs at the stage of synaptic partner matching. As can be seen from these figures, there is a large number of differentially expressed cell-surface molecules. We only have limited knowledge about the ligand-receptor pairs among these proteins; in most cases, we don’t know these interactions are attractive or repulsive. It will take substantial amount of additional work to test the functional significance of these expression data. Nevertheless, these expression data have provided a critical first step to tackle this important problem in developmental neurobiology.

4) A small number of OR genes shows co-expression with other OR genes are expressed in more than one class of ORNs. Is there anything in common among the OR genes that are co-expressed. It is clear that some are recent duplication events where the genes are repeated in a single genomic locus (Or65a/b/c or Or22a/b).But others are not so close on the genome. Is there anything in common among these ORs? For example, is there any amino acid sequence similarity suggestive of more distant duplication and translocation of the gene to another region in the genome that retained some of the same regulatory elements? Or co-expression seen in ORNs found in the same sensilla type, which might indicate regulatory similarities?

We would like to clarify a few different cases regarding co-expression. (a) It has been well documented in the literature that certain *Or* genes, such as *Or22a* and *Or22b*, are co-expressed, likely because of common regulatory elements. (b) Our transcriptome data have supported previous documented co-expression of *Ors*. (c) While our paper is in review, two preprints came out showing substantially more widespread co-expression of sensory receptors in the olfactory system of flies and mosquitos based on co-receptors (Orco, Ir25a, etc.). We did not draw a lot of attention to this co-expression in the original manuscript, and in light of the reviewer’s comments and the preprints, we have expanded this analysis, including citing previous studies in (a) and our support for those; document systematically new co-expression from our own data (b), and refer to the two new studies. Please see Major Addition 2 above for more details.

5) The authors make the conclusion that a given OR is expressed in the same ORN from birth until adulthood, in contrast to mammalian ORNs which initially express multiple OR genes but then consolidate to express a single one. They use both their RNAseq data and a permanent genetic labeling to show this. In the text they mention that generally most of the ORN that express a single OR gene innervate a single glomerulus even when they perform the permanent genetic labeling. However, in Figures 3E and the supplement, they show images where permanent labeling leads to innervation of some weak secondary glomeruli by a single OR-GAL4. I could not find an explanation for this in the main text. Can the authors elaborate on these? Is this a GAL4 artifact? Or a real indication that these OR genes initially are also expressed in other ORNs but are later consolidated to a single ORN class?

This is a good point and we should have clarified this better in the text. There are ~10% examined antennal lobes showing weak additional expression using permanent labeling, meaning that in ~90% cases each *Or* is only expressed in one ORN type from early development. (By contrast, in the mouse olfactory system, almost all immature ORNs expresses multiple olfactory receptor genes.) Even in the antennal lobes with additional labeling, the labeling can originate from a combination of ORNs, olfactory projection neurons (PNs), and local interneurons. We also notice that these additional labeling are mostly asymmetric, further suggesting they are not from ORNs because most ORNs targeting their axons to both antenna lobes in the brain while PNs are mostly targeting to one side. Thus, our data suggest in vast majority of cases, individual *Or* genes only express in a single ORN types. We have clarified this in the text and have provided additional supplemental images depicting this (Figure 3—figure supplement 1 E–G).

6) It would be good to get an expanded clustered heat maps and lists for all ORN classes and the expression of specific gene groups, like cell surface receptors, transcription factors, etc.

Thank you for this great suggestion. See Major Addition 1 above for details.

7) Some of the wording on the figures, especially in heatmaps etc. are extremely small. Please make them bigger in addition to providing a separate table with expression levels.

We have amended that and have set our minimum font size to 6 at the final magnification.

8) For Figure 1B, can you provide the names of the ORs co-expressed that are shown on the heatmap?

We presume that the reviewer meant Figure 3B. We have added names next to the wheel in Figure 3C, which has more space.

Reviewer #2:In this work, the authors provide a thorough analysis of single cell sequencing from developing *Drosophila* ORN tissue. This is the first comprehensive single cell RNA seq analysis from multiple developmental stages of ORN development and undoubtedly a valuable resource for the field. Overall the paper is well written and the experiments are rigorous. However, I have some significant concerns.1) Novelty: While this is the first single-cell RNA seq paper, it is not the first to conduct RNA-seq on ORN tissue across development. Notably, these studies are not cited and the results are therefore not compared.

Regarding citation to previous bulk RNAseq paper, thank you very much for pointing out this important deficiency. Please Major Addition 3 above for details. Single-cell RNA-seq studies provide additional value over bulk RNA-seq of the entire third antennal segment: we are identifying genes that are expressed in receptor neurons, and more importantly, those that are differentially expressed in distinct types of receptor neurons. We regret that such significance did not come out in our original manuscript. We hope that the reviewer will agree that by including more extensive analyses of differentially expressed genes described in Major Point 1 described above, and comparisons with the bulk RNAseq data in Major Point 3 described above, we have made these advances more apparent.

2) There are very few new conclusions to be drawn from this paper that haven't been identified in other studies. It was already well known that ORNs only express one receptor in their lifetime unlike mice. The few genes that are novel to ORN development are not discussed in any detail. Only forkhead gets a passing mention in the text

See point 1 above regarding new conclusions. We respectively disagree with the notion that “It was already well known that ORNs only express one receptor in their lifetime unlike mice.” Perhaps it is a widely held belief, but we are not aware of previous experiments that rigorously demonstrate this. Indeed, in the mouse, a previously widely held belief had been that each ORN expresses a single odorant receptor throughout development also, until single-cell RNA-seq study demonstrated that transient expression of multiple Or genes occurs during development (Hanchate et al., 2015; Tan et al., 2015). The data we presented in Figure 3 showed for the first time that at least for the ORN types we examined, transient expression of multiple receptors does not occur in *Drosophila*.

3) Even though few novel genes are mentioned, there is also almost no discussion of known genes either. Do the authors see expression of wiring genes like robo, dscam, and DIP/Dprs? What about TFs known to control ORN development? Atonal, rotund, amos, ect.? There are several TFs known to maintain OR expression in the adult antenna. Do the authors see expression that matches these results? Again, several studies are not cited here.

These have been amended—see Major Additions 1 and 3 above.

Reviewer #3:This study generated single-cell transcriptomes from neurons of the *Drosophila* antenna at 24h APF and adult antennae. It uses computational techniques to match most (60% and 80%, respectively) transcriptional clusters to defined neuronal types. The authors have earlier published (Li et al., 2020) single-cell transcriptomes at 42h APF, and those data are compared to the new data in the current study. It's not clear that the present study produces a major conceptual advance, but it describes the application of single nucleus sequencing to *Drosophila* and provides a resource that should be very useful to many in the field.1) Do ORNs in the same sensillum (for example ab2A and ab2B) share more similar transcriptomes?

In general, yes. In our adult data, there are a few instances where we observe 2–3 ORN types comprising one transcriptomic cluster. Some of these examples are cases where these ORN types are from the same sensillum, indicating that their transcriptomes are very similar to each other’s. For example, VL1 and VM1 ORNs derived from the ac1 sensillum are found in the same cluster and so are DM3- and VA6-projecting ORNs (ab5 sensillum). The fact that we cannot separate these ORN types at the cluster level indicates that their transcriptomes are very similar at this stage. We have clarified this in the text.

In Figure 7 (panels B and C), we have specifically looked at transcriptome similarity of all annotated ORNs at each stage. We initially had multiple other panels in this figure, but we removed some as to not overwhelm the reader and to more clearly state our findings. Specifically, we found that while some ORNs found in the same sensillum, like VA1v- and VA1d-projecting ORNs at 24h APF, have the most similar transcriptomes to each other, this is not the case for all ORNs at both stages. For instance, at both 24h APF and the adult stage, V ORNs are not most similar to other ORNs (VA2/DM1) within the ab1 sensillum. Similarly, at 24h APF, ORNs from the ac4 sensillum (VM4 and VL2a) do not share more similar transcriptomes with each other but rather are more similar to ORNs taking the same axonal trajectory. We have clarified this in the text as well.

2) It would be helpful to compare the results of snRNAseq with results from previous RNAseq of the entire antenna. This comparison might reveal if there is any bias imposed by the snRNAseq methods used here. Are fewer Ors identified in the present study? If so, are there any commonalities among those that are missing from the snRNA seq? Are there striking differences in the relative abundances between the two methods? Was Or85c observed in whole antennal RNAseq?

Great point. See Major Addition 3 above.

3) Figure 1M: Why does the gene ontology analysis identify "chemotaxis" as a top term at 24h but not at the adult stage?

We assume that the reviewer meant Figure 2M. In Gene Ontology term, “chemotaxis” refers to “The directed movement of a motile cell or organism, or the directed growth of a cell guided by a specific chemical concentration gradient. Movement may be towards a higher concentration (positive chemotaxis) or towards a lower concentration (negative chemotaxis).” In principle, it should include cell migration, axon guidance, as well as organismal chemotaxis. Perhaps in multicellular organisms, this term is dominated by cellular rather than organismal chemotaxis. Additionally, organismal chemotaxis in response to a chemical cue would likely fall into the GO category “chemosensory behavior”, which is displayed in the adult GO terms.

4) The authors should clarify here the limitations of the analysis. "Cluster-specific Or and Gr expression allowed us to match 13 transcriptomic clusters with their glomerular types." There should be a clearer explanation here for why the number is not higher. Likewise, the authors should explain more explicitly why transcriptomes of some ORNs are indistinguishable at the adult stage – they should express different receptors. It would be helpful to provide a list of clusters containing multiple neurons, indicating for each cluster which neurons map to it.

Our first paragraph of Discussion gave three explanations for this phenomenon. We have referred to Discussion so readers know that explanation is forthcoming.

5) The studies of the requirements for G proteins (for example Kain et al.) need to be stated more accurately.

Thank you for drawing our attention to this. We have completely removed this section from the paper and replaced it with a section evaluating co-receptor expression in our data (new Figure 9). Instead of comparing Or and Ir transcriptomes as we had previously done, we compared all of the annotated adult transcriptomes and identified type-specific neuronal signaling genes in Figure 10.

6) Figure 3C. The chord plot shows co-expression of Or85c and Or85b, which are located near each other in the genome. Is there a tendency for other genes that are near each other to be co-expressed?

See reviewer 1 point 4 and Major Addition 3 above.

7) The detection of mitochondrial transcripts in ORN nuclei is said to result from dissociation artifacts. Obps are found here to be broadly detected in ORN nuclei. Obps are expressed at extremely high levels in accessory cells. Could their detection in ORN nuclei be artifactual as well?

Thank you for this excellent point! To address the reviewer’s point, we have compared expression of *Obps* in ORNs to the published *Obp* expression in whole antennal bulk RNA-seq (Menuz et al., 2014), and noticed that many of the highest expressed *Obps* in the whole antenna sequencing were also the highest expressed in our dataset. The plot below shows Obp expression in our dataset [color scale: log_2_(1 + transcript counts per million)] rank ordered by the expression levels reported by Menuz et al. (2014) from high to low (y-axis). It is apparent that the higher the bulk RNA-seq levels, the more expression we detected in our dataset. Thus, we think that because *Obps* are so highly expressed in accessory cells (at least one order of magnitude higher than the abundant *Orco* gene in bulk RNA-seq data), it is likely that the *Obp* detection in our nuclei is the result of artifacts caused by ambient RNA associated with nuclei during the dissociation process. We have included a new panel (Figure 2—figure supplement 1O) and discussed this caveat in the legend and in the text associated with Figure 2. We also re-did the GO and STRING analysis for Figure 2 after removing Obp expression from the adult transcriptomes. We are grateful to reviewer 3 for pointing this out.

8) Please state more explicitly how many single nuclei, and how many single cells, are discarded from the analysis, and the reasons for discarding them.

We have described the parameters for including a cell/nucleus in our analysis in the text associated with Figure 1 and the Materials and methods section. The parameters are also highlighted in Figure 1—figure supplement 1. We have now also added the total number of cells sequenced to the Materials and methods section.

9) There is some concern about the use of intronic reads. In Figure 5C, for example, Ir41a is reported to be expressed in many neurons. Ir41a is a very long gene. Could its apparent expression represent noise resulting from using intronic reads?

Thank you for drawing our attention to this. Since nuclei only contain a fraction of the total transcripts (many are pre-mRNA with introns) in cells, the inclusion of intronic reads to increase gene detection is standard for single-nucleus RNA sequencing and aids in high-resolution cell type identification from snRNA-seq data in a large number of studies published in the past few years (Bakken et al., 2018, https://doi.org/10.1371/journal. pone.0209648; Thrupp et al., 2020, https://doi.org/10.1016/j.celrep.2020.108189; Cui et al., 2020, https://doi.org/10.1016/j.devcel.2020.02.019; Kebschull et al., 2020, https://doi.org/10.1126/science.abd5059; Slyper et al., 2020, https://doi.org/10.1038/s41591-020-0844-1). And we have adopted this standard in our own analysis and showed the advantages of including introns in our side-by-side comparisons detailed in Figure 2.

In general, longer genes have a higher chance of creating the appearance of “noise” in the data, whether it is long introns or long exons. The appearance of “noise” in singlecell/single-nucleus RNA sequencing data can occur for several reasons. First, other annotated genes fall within the intronic/exonic region(s) of the gene of interest. The settings on our current sequencing reads alignment/counting method should not count reads that contain portions of multiple annotated genes. Specifically, for the *Ir41a* genomic locus, our analysis did not reveal any annotated genes within it. Thus, this is unlikely to be the source of noise in our data. Second, our current normalization method, which uses counts per million reads (CPM) and is a standard normalization method in single-cell sequencing, can lead to more sequencing read counts of a single copy of a longer gene’s transcript than a shorter gene. (Unlike bulk RNA sequencing, reads are not typically normalized to a gene’s length in single-cell/single-nucleus RNA-seq protocols because normalization to a gene’s length can introduce other artifacts.) Third, we used an oligo-dT based primer to generate cDNA from poly-adenylated mRNA during library preparation. If a gene has long poly(A) stretches, non-specific internal poly(A) priming can occur which may lead to more copies of cDNA being generated for that gene. The Smart-seq2 platform (the platform we used in this study) does not use unique molecular identifiers (UMIs) to tag individual mRNA molecules and, hence, enable the discrimination between original mRNA molecules and duplicates generated during cDNA amplification (Ziegenhain et al., 2017, https://doi.org/10.1016/j.molcel.2017.01.023). Thus, since *Ir41a* is much longer than other antennal sensory receptors it is possible that the very low-level detection of this transcript in multiple cell-types could have originated from one of the latter two explanations. Since we wanted to adhere to standard practices in Smart-seq2based single-nucleus sequencing, we avoided both normalizing *Ir41a* (or any of the other genes in this study) to its length and removing “noisy” reads from the *Ir41a* counts. Instead, we added an explanation for the low-level expression of *Ir41a* (and *Ir93a,* which is also longer than many other antennal sensory receptors and displays broad low-level expression) in many cells to the legend associated with Figure 5.